



# Quantifying the added value of high resolution climate models: A systematic comparison of WRF simulations for complex Himalayan terrain

Ramchandra Karki [1,2], Shabeh ul Hasson [1,3], Lars Gerlitz[4], Udo Schickhoff [1], Thomas Scholten[5], Jürgen Böhner [1]

[1]  Center for Earth System Research and Sustainability, Institute of Geography, University of Hamburg, Bundesstraße 55, 20146 Hamburg, Germany
[2]  Department of Hydrology and Meteorology, Government of Nepal, 406 Naxal, Kathmandu, Nepal
[3]  Department of Space Sciences, Institute of Space Technology, Islamabad 44000, Pakistan
[4]  Section Hydrology, GFZ German Research Centre for Geosciences, Telegrafenberg, 14473 Potsdam, Germany
[5]  Soil Science and Geomorphology, University of Tübingen, Department of Geosciences, Rümelinstrasse 19-23, 72070 Tübingen, Germany
Correspondence: Ramchandra.Karki@studium.uni-hamburg.de

**Abstract.** Mesoscale dynamical refinements of global climate models or atmospheric reanalysis have shown their potential to resolve the intricate atmospheric processes, their land surface interactions, and subsequently, realistic distribution of climatic fields in complex terrains. Given that such potential is yet to be explored within
the central Himalayan region of Nepal, we investigate the skill of the Weather Research and Forecasting (WRF) model with different spatial resolutions in reproducing the spatial, seasonal and diurnal characteristics of the near-surface air temperature and precipitation, as well as, the spatial shifts in the diurnal monsoonal precipitation peak over the Khumbu (Everest), Rolwaling and adjacent southern areas. Therefore, the ERA-Interim (0.75º) reanalysis has been dynamically refined to 25, 5 and 1 km (D1, D2 and D3) for one complete
hydrological year (Oct 2014 – Sep 2015), using the one-way nested WRF model run with mild-nudging and parameterized convection for the outer but explicitly resolved convection for the inner domains. Our results suggest that D3 realistically reproduces the monsoonal precipitation, as compared to its underestimation by D1 but overestimation by D2. All three resolutions however overestimate precipitation from the westerly disturbances, owing to simulating anomalously higher intensity of few intermittent events. Temperatures are
though generally well reproduced by all resolutions, winter and pre-monsoon seasons feature a high cold bias for high elevations while lower show a simultaneous warm bias. Contrary to higher resolutions, D1 fails to realistically reproduce the regional-scale nocturnal monsoonal peak precipitation observed at the Himalayan foothills and its diurnal shift towards high elevations, whereas D2 resolves these characteristics but exhibits a limited skill in reproducing such peak at the river valley scale due to the limited representation of the narrow
valleys at 5 km resolution. Nonetheless, featuring a substantial skill over D1 and D2, D3 simulates almost realistic shapes of the seasonal and diurnal precipitation and the peak timings even at valley scales. These findings clearly suggest an added value of the convective scale resolutions in realistically resolving the topo-climates over the central Himalaya, which in turn allow simulating their interactions with the synoptic scale weather systems prevailing over High Asia.

**Keywords:** WRF, Nepal, Himalayas, temperature, precipitation, diurnal cycle, seasonal cycle, valley





## 1. Background

Featuring a complex terrain along the 2500 km arc, the Himalaya barrier extensively contributes in redefining the regional climate: 1) by controlling the cold and dry air advection from central Asia, and; 2) by enhancing and redistributing the incident precipitation (Norris et al., 2015). Hence, the Himalayas play a crucial role in defining the regions' hydrometeorology and hydrology, which in turn ensures the food and water security of 1.6 billion people and the sustainable development of the agrarian economies downstream.

Since Nepal is situated in the center of the Himalayan arc, its climate also results from a sophisticated interaction between the steep and complex Himalayan terrain and the two large scale circulation modes: the south Asian summer monsoon and the high-level baroclinic westerlies. The hydro-climatology of Nepal is dominated by the south Asian summer monsoon that provides more than 80 % of the total annual precipitation, as winter is mostly dry and cold (Shrestha, 2000; Karki et al., 2017). Being monsoon-dominated mountainous region, Nepal is highly susceptible to the developments of heavy rainfalls that trigger hydro-meteorological disasters, threaten the food and water security and cause loss of life. Such problems of food and water security and disaster management in Nepal and Himalaya seem to be further exacerbated by the observed warming and its altitude-dependent intensification, thinning and retreat of glaciers, formation and expansion of glacial lakes, shift of treeline to higher elevations, changes in precipitation seasonality and increasing frequency and severity of floods and droughts (Karki et al., 2010; Soncini et al., 2016; Schickhoff et al., 2015; Hasson et al., 2016a; Karki et al., 2017). Such vulnerability and exposure of the mountainous communities to the adverse socio-economic impacts of changing climate and its induced disasters greatly emphasize on understanding their drivers and their subsequent impacts on various sectors of life for devising local scale adaptation strategies. This requires, before all, a fine-scale climatic information, particularly of surface air temperature (hereinafter temperature) and precipitation, which are also amongst the essential inputs to impact assessments models (Böhner and Lehmkuhl, 2005; Karmacharya et al., 2016; Gerlitz et al., 2016).

However, meteorological observations are too sparse in the region while spatially complete gridded (interpolated or remotely-sensed) observations, reanalysis products and regional-to-global scale climate models' datasets are too coarse to represent the topo-climates over the deep and broad river valleys, and over the steep slopes and ridges of the complex Himalayan terrain (Lang and Barros, 2004; Palazzi et al., 2013; Gerlitz et al., 2016; Hasson et al., 2016c). Consequently, interaction of the distinct synoptic to mesoscale circulations with local topo-climates is yet understood only indistinctly. Thus, a better understanding of the local atmospheric processes and their interaction with prevailing circulation modes over the complex Himalayan terrain largely depends upon fine scale climate model simulations featuring a satisfactory representation of the present climate (Hasson et al., 2016b), usually obtained through dynamical downscaling of reanalysis products.

The WRF model is a proven tool to dynamically downscale at very high-resolutions (<10 km) in a complex mountainous terrains (Rasmussen et al., 2011; Viale et al., 2012; Jiménez and Dudhia, 2013). Unlike temperature, WRF does not necessarily show an improvement for precipitation when going to high resolutions, and there are regional differences. For example, over the Tibetan plateau (TP) Maussian et al. (2014) have shown that the spatial patterns, interannual variability and seasonality of precipitation are well reproduced by WRF simulations performed at 30 and 10 km, with the latter one showing relative improvements. But over the central Himalaya – where topography is relatively complex as compared to TP – their 10 km simulation suggests an underestimation of monsoonal precipitation by half. On the contrary, over the central Himalaya, the 6.7 km convection permitting WRF simulation by Norris et al. (2016) suggests an overestimation up to twice the of observed monsoonal precipitation. Focusing on the small Langtang catchment in Nepal, Collier and



Immerzeel (2015) illustrated even poor performance of 5 km convection permitting WRF simulations featuring higher underestimation of monsoonal precipitation compared to 25 km resolution simulations with parameterized convection. Such regionally distinct WRF results could be due to the extremely heterogeneous terrain and varied micro-climates of the Himalayas.

The diurnal distribution of precipitation is another important characteristic of the monsoonal precipitation, as it controls the circulation characteristics (latent heat release) and affects the precipitation magnitude (Sato et al., 2008; Bhatt et al., 2014; Shrestha and Desar, 2014). Hence, a better imitation of the diurnal cycle characteristics by the climate models can ensure their realistic representation of the monsoonal precipitation and particularly their topo-climatic driving factors. However, in coarser horizontal resolutions (>15 km), WRF has shown a

deficiency in simulating even large scale mid-night to early-morning peak precipitation, evident at the Himalayan foothills (Jyoti et al., 2012). A 12 km WRF run though broadly reproduces the principle shape of the diurnal cycle but fails to capture the lull precipitation hour, the timing of the diurnal peak and its southward migration (Bhatt et al., 2014). Besides horizontal resolution, convection parameterizations can also greatly affect the climate model's performance in reproducing the distributional characteristics of diurnal precipitation.

Investigating four different model resolutions (28, 14, 7 and 3.5 km) over Tibetan Plateau for April, Sato et al. (2008) demonstrated that only high resolution (<7 km) and convection permitting (non-parameterized convection) runs reproduce proper development of local afternoon clouds and diurnal features. Norris et al. (2016) also showcased a relative improvement of 6.7km convection-permitting WRF simulation over the coarser and parameterized-convection simulation in reproducing the nocturnal precipitation magnitude.

Concurrently, their simulation failed to resolve the nocturnal peak, evident in the mountainous river valleys, such as, Khumbu. Collier and Immerzeel (2015) evaluated the WRF simulation at 1 km resolution over very small area, however they did not analyze the spatial pattern of diurnal precipitation. Thus, it is still unclear what added value a kilometer scale convection permitting simulation may have in terms of reproducing the characteristics of diurnal cycle of precipitation over a complex Nepalese Himalayan terrain.

Mostly focusing on the monsoonal precipitation, the above studies are constrained by missing either: 1) a fully explicit cloud resolving scale (<5 km), or; 2) the assessment of impacts of high resolutions and explicitly resolved convections. Additionally, none of the studies extensively evaluated the WRF simulation against a large number of observations from different altitudinal ranges of the central Himalaya. Against this background, this study performs a systematic evaluation of one-year WRF simulation (Oct 2014 - Sep 2015) over highly

complex terrain of Khumbu (Everest) and Rolwaling Himal in terms of reproducing the spatial distribution and annual and diurnal cycles of both temperature and precipitation, and for the latter, additionally the shift in timings of the peak precipitation hour. Further, the added value of two fine-scaled (at 5 and 1 km) convection permitting simulations was assessed relative to coarser 25 km simulation with parameterized convection, based on a dense meteorological network, including the novel observations from our TREELINE project.

The remainder of the paper is organized as follows: section 2 describes the study area while Section 3 explains the model configuration, observations and data analysis. Section 4 reports the WRF skill against the station observations for precipitation and temperature, where seasonal error statistics and seasonal and diurnal cycle characteristics are discussed at first, then the capability of adopted WRF resolutions in reproducing the spatial distribution of surface variables, their altitudinal gradients and the interaction of local mountain valley-wind and

large-scale circulation (responsible for monsoonal diurnal precipitation characteristics) are presented. Comparison of WRF skill across the seasons and the adopted resolutions is particularly focused. Section 5 concludes the major findings.



## 2. Study area

The target region of the presented study spans over the physiogeographic regions of middle-mountains and the high-Himalayas of Nepal, and features an elevation range starting from 400 masl to the world's highest peak at 8848 masl (Mt. Everest). The study area contains many glaciers and glacial lakes, which are the main source of perennial freshwater supply for the downstream regions. Detailed analysis is carried out for our WRF domain D3, which covers mainly the Dudhkoshi and Tamakoshi sub-basins of Koshi river basin and surroundings, located in the northeast Nepalese Himalayas (Figure 1b). Our primary focus is on the upstream sub-basin of Dudhkoshi at Khumbu and, Rolwaling which drains to the west to Tamakoshi River.

The climate of the region varies from temperate with dry winters and hot summers in the southern river valleys to polar in the high Himalayas (Karki et al., 2016). There are mainly four typical climatic seasons: winter (December to February), pre-monsoon (March to May), monsoon (June to September), and post-monsoon (October to November). Winter precipitation mainly occurs due to extra tropical cyclones, called western disturbances, bringing moisture from the Mediterranean, Arabian or Caspian Seas (Cannon et al., 2014; Böhner et al., 2015). Norris et al. (2015) also illustrated the west and southwest cross-barrier mid-tropospheric moisture flow as the dominant factor (mechanical driver) for winter snowfall in Himalayas. Winter precipitation is mostly received in the form of snow at high-altitudes, which though, limited in volume, is vital for winter crops as well as for regulating the streamflow during hot and dry pre-monsoon season (Böhner et al., 2015). Pre-monsoon season is characterized by strong solar insolation, which results in maximum temperatures. Pre-monsoonal precipitation, usually observed in the evening, is generated primarily by localized convective instability with heating) and is supported by the mountain uplift in the areas of surplus moisture. Whereas, post-monsoon is the driest season over the study region. The synoptic scale western disturbances occasionally influence both early pre-monsoon and late post-monsoon precipitation, causing snowfall at high altitudes.

Monsoonal precipitation is characterized by strong southerly and southeasterly flow from the Bay of Bengal (Böhner, 2006; Hasson et al., 2016a&b) and contributes to more than 80 % of the annual precipitation (Wagnon et al., 2013; Salerno et al. 2015; Gerlitz et al., 2016; Karki et al., 2017). Within the Khumbu region, monsoon precipitation decreases along main river valleys between elevations of 2800 to 4500 masl, however, precipitation sums around peaks and ridges are about 4-5 times larger than at valley bottoms (Higuchi et al., 1982).

The peak annual precipitation is observed around 2500 masl for the Koshi basin and sharply decreases above this height (Salerno et al., 2015). At the southern slopes of Mt. Everest (8000 masl), mean daily temperatures vary between -42 and 4 °C (Gerlitz, 2014) while lower river valleys are very hot. Other common meteorological features are locally generated day-time up-valley and night-time down-valley winds (Ueno et al., 2008; Böhner et al., 2015; Gerlitz et al., 2016).

## 3. Methods

### 3.1 Model Configuration

WRF is a fully-compressible non-hydrostatic model that offers a large number of physical parameterizations and is suitable for simulating across a variety of horizontal and vertical scales. We have setup the WRF version 3.6.1 for the concurrent run of three one-way nested domains D1, D2 and D3 configured at 25, 5 and 1 km horizontal





grid spacing, respectively (Figure 1 a). D1 spans over most of the Bay of Bengal and parts of the Arabian Sea (16.0-39.9 °N, 66.6-99.9 °E), which are the main moisture sources of prevailing synoptic weather systems. The D2 covers more than the eastern half of Nepal along the east-west extent (83.8-88.9 °E) and includes the Tibetan Plateau in the north and the Endo-gangetic plain in the south (25.6-30.0 °N). D3 covers (27.2-28.3 °N,

85.7-87.1 °E) mainly the Dudhkoshi basin including the Khumbu sub-basin, the Tamakoshi basin and the Rolwaling sub-basin, where stations with hourly observations are located. The real topography is better represented in the high-resolution domains, particularly in D3 with mostly offset within ±400 m (Figure 1 b and Supplement Figure S1). The vertical resolution of WRF was set to 50 levels (from 20 m height to 50 hPa) with 11 lowest levels lying below the height of 1 km (Table 1).

The physical parameterization schemes include the Morrison microphysics scheme (most sophisticated double moment) and the planetary boundary layer scheme of MYNN level 2.5. The Kain-Fritsch cumulus parameterization option is used for D1 but turned off for D2 and D3 in order to explicitly resolve the precipitation processes. The setup uses the latest Noah multi-parameterization land surface model (MP LSM) (Niu et al., 2011) as it better represents the terrestrial moisture processes than the Noah LSM, and relies on the

shortwave and longwave radiation schemes of the NCAR community atmospheric model (Collins et al., 2004, Norris et al., 2016). The selected physics options are consistent with Collier and Immerzeel (2015), who reported improved simulation of the surface variables over the Himalayan region. Initial and lateral boundary conditions for D1 were derived from the ERA-Interim reanalysis, which has the spatial resolution of 0.75° × 0.75° and the vertical resolution of 38 pressure levels (Dee et al. 2011). Both sea surface temperature (SST) and

lateral boundary conditions were updated at six hours interval.

Mesoscale climate models may have difficulties in representing the large-scale features (Jones et al., 1995) which can be avoided through nudging the model (Von Storch et al., 2000). Since nudging can controls the freedom of downscaling model to deviate from the driving models (Alexandru et al., 2008), we have applied grid analysis nudging only to D1 and merely for the horizontal winds, potential temperature, and water vapor

mixing ratio in the vertical levels above the planetary boundary layer and above the lowest 15 model levels. Previous studies have demonstrated the improved simulation of mean and extremes of surface variables with the application of such mild nudging (Otte et al., 2012; Collier and Immerzeel, 2015).

The WRF run was performed for one complete hydrological year with two initializations, in order to optimize the available computational resources. The first initialization was performed at 0000UTC 15 September 2014

(ending at 0000UTC 1 June 2015), while the second initialization was performed at 0000UTC 15 April 2015 (ending at 0000UTC 1 Oct 2015). The first 15 days from the first run with wet period initialization (Collier and Immerzeel, 2015) and first 45 days from the second run with dry period initialization (Norris et al. 2016) were regarded as the model spin-up time to ensure the equilibrium between the boundary conditions and the model dynamics (Argüeso et al., 2012) that yielded one-year simulation from October 01, 2014 to September 30, 2015.


### 3.2 Observations

In view of the complex Nepalese Himalayan terrain where the surface variables vary extensively within a short time and space, robust high-resolution observations are necessary for the detailed evaluation of the fine-scaled simulation (Hasson et al., 216b). This is why the focus of D3 was set to the area hosting a relatively dense

network of 29 meteorological stations, which are being maintained by four different organizations (Figure 1b,



Table 2). Within the Rolwaling, seven stations collecting 15-minute temperature and precipitation observations are being operated by the Institute of Geography, University of Hamburg, Hamburg under the TREELINE project that aims to understand the response of undisturbed treeline ecotone to the global warming (Gerlitz et al., 2016; Schwab et al., 2016; Müller et al. 2016a). In the Khumbu region, 1-minute observations from five Ev-K2-

CNR stations (www.evk2cnr.org) and 15-minute observations from Lukla station of the Department of Hydrology and Meteorology (DHM) are available. Further, in Khumbu valley and surroundings, three stations from the Laboratoire Hydrosciences, CNRS, Institute of research for development (IRD), Université de Montpellier, France, provide daily observations of temperature and precipitation. In the southern hills, the DHM station of Okhaldhunga provides 15-minute observations of both temperature and precipitation, while in the

lower altitudes eight DHM stations provide daily observations of precipitation only. Similarly in the lower reaches of the Rolwaling, four DHM stations provide 15-minute precipitation. We have obtained all the available hourly and daily temperature and precipitation from the stations of the study area (Table 2).

It is worth noting that the type of precipitation gauge in available observational networks is different, such as, manual US standard gauge, tipping bucket, and weighing gauge. Most of the automatic weather stations use

tipping buckets without rim heating, while the manual stations use the manual US standard gauges. Relative to manual gauges, tipping buckets are known for their underestimation of rainfall by 10% (Talchabhadel et al., 2016) and for even higher underestimation of snowfall. These gauges yield incorrect timings of the precipitation events as the solid precipitation during night or at sub-zero temperature may be deposited in the upper funnel (Karki, 2012) and only contribute to the measurements after snow melting due to strong solar insolation

(Shrestha et al., 2011). Further, the snowfall measurements are affected by the rim snow capping, snow deposition in the upper funnel and overflow in case of heavy precipitation events. Goodison et al. (1998) have also pointed out that commonly used precipitation gauges strongly underestimate the solid precipitation. In view of these limitations, the visual inspection of the datasets and consistency check between nearby stations within the network was performed. The value crossing general extreme (e.g. negative and >500 mm/day for

precipitation, and < -40 and > 50 ºC for temperature) and plausible station limits, constant value of temperature for whole day, without precipitation for whole monsoon season and abrupt precipitation (>30mm/hour) in one station but without precipitation in nearby stations (around 5km) were considered as erroneous data set and excluded from analysis.

### 3.3 Model validation and data analysis

Before validating the simulated surface variables, we analyze the model skill in simulating the seasonal contrasts of dynamic and thermodynamic variables, particularly for D3. Therefore, area averaged annual cycles of low and upper level zonal winds, low level temperature and specific humidity, precipitable water, outgoing long wave radiation (OLR) and height of the planetary boundary layer (PBL) are plotted (Figure 2).

Subsequently, we evaluate the WRF simulations considering the above mentioned point observations and

compare them with the collocating grid point of the particular domain. The differences between the real station elevation and the associated grid cell of the WRF model, arising from the smoothed terrain of the WRF domains (Supplement Figure S1), might lead to a systematic bias of the modelling results. Thus, simulated temperatures are adjusted using a constant lapse rate of 6 ºCkm-1, which has been observed over the whole Koshi basin (Salerno et al., 2015). Since precipitation is highly variable in space and time, it has not been corrected. The

periods of observed data gaps were ignored for the analysis.



For daily-to-monthly temperature validation, statistical measures of bias (WRF-Observation), mean absolute error (MAE), root mean square error (RMSE), ratio of RMSE and standard deviation of observations (RAR) and correlation (r) are applied to the all-stations pooled data (not all stations average). This pooling procedure, considering all stations data point, is expected to better account for the local-scale features in statistical analysis

than averaging which smooth out the overall series (Soares et al., 2012). Similar measures were applied to precipitation, however instead of Bias and MAE, the normalized percentage bias (NBias) and the mean absolute percentage error (MAPE) were calculated using Eqn. 1 and 2, which better handle the large number of data points with zero precipitation.

$$NBias = \frac{100}{n} \sum_{m=1}^{n} \frac{(P_m - O)}{O} \qquad (1)$$

$$MAPE = \frac{100}{n} \sum_{m=1}^{n} \frac{|(P_m - O)|}{O} \qquad (2)$$

where $P_m$ refers to the WRF simulated precipitation, O refers to the mean observed precipitation and n indicates the total number of data points. For seasonal scale, standard normalized precipitation bias - defined as the ratio of bias and observation - is also calculated. Further, the WRF skill in simulating high intensity precipitation and extreme temperatures is analysed for each season by means of a percentile value comparison, given the fact, that

these events have huge socio-economic and environmental consequences for the study region.

The spatially complete WRF simulations with different spatial resolution are analyzed in order to derive altitudinal gradients of surface variables. This enables the assessment of seasonal lapse rates of temperature and elevational profiles of precipitation, which are frequently required for climate impact investigations.

As the diurnal precipitation variability controls the surface energy budget (latent heat release) and water balance

(Sato et al., 2008; Bhatt et al., 2014), we also analyze the diurnal precipitation characteristics. However, in view of the deficiency of tipping bucket gauges which frequently observe an artificial precipitation peak in the morning instead of during the night, we restrict our analysis to the monsoon season and to stations below elevations of 5000 masl. In order to investigate the topo-climatic processes, leading to a spatial differentiation of precipitation, we grouped the observations into three discrete classes. All stations located near the valley bottom

are grouped into upper and lower valley stations based on an elevation-threshold of 2500 masl. The stations that are located along the ridges are classified as ridge stations. We assess the WRF skill in reproducing the observed peak precipitation timing over the lower and upper valleys as well as over the mountain ridge. Besides the peak precipitation timing, we further analyze its propagation characteristics from high mountains towards the river valleys and the southern foot hills (for D2 and D1) from afternoon/late-evening to mid-night/early morning.

Concluding, the mountain valley circulation and its representation in the WRF model is investigated and its interplay with spatial and diurnal precipitation characteristics is illustrated in detail.

**4. Results and Discussions**

**4.1 Model validation results**

The model satisfactorily reproduces the seasonal contrasts of dynamic and thermodynamic variables. Monsoon season is characterized by moist atmospheric conditions (high specific humidity, precipitable water, and relatively low OLR), warmest temperatures and minimal upper and lower level westerly flow. Further the PBL height is significantly reduced as a result of high latent but low sensible heat fluxes. During post-monsoon, a reversal of surface and upper air winds, a decrease of temperature and moisture as well as an increase of OLR





and PBL height are simulated. Winter is fully dominated by strong westerly surface and upper airflow as well as dry and cold atmospheric conditions. Similarly, during pre-monsoon, higher temperatures and reduced moisture supply cause dry soil conditions, which in turn yield smaller latent but larger sensible heat fluxes, resulting in a deeper PBL.

The comparison of the near surface meteorological variables of the WRF simulations with the station observation in the target region revealed the following results:

**Precipitation**

Observations show that the complex terrain of the region yields to a highly heterogeneous precipitation
distribution (from 400 to 2700 mm/annum). In general, river valleys are found drier than the surrounding mountain ranges, however a majority of precipitation falls below elevations of approximately 3500 masl (exponential decrease of saturation vapour pressure with altitude). Hence, high altitude areas and leeward slopes of high mountain ranges receive little precipitation (e.g. upstream stations of Khumbu and Rolwaling, which are described in detail in section 4.2). During winter and post-monsoon season only few precipitation events are
observed, these contribute to ~6% of the annual precipitation sum. From late pre-monsoon to late-monsoon season precipitation is permanently high and contributes to ~16 and ~78% of the annual precipitation, respectively, indicating dominance of monsoonal precipitation and induced disasters (Figure 3).

The seasonal error statistics of simulated daily and monthly precipitation (calculated against station observations) suggest distinct skills of each model domain though varying across seasons (Table 3). For
monsoon, D1 features roughly 50% negative bias whereas D2 overestimates precipitation by up to 40%. D3 also overestimates the monsoonal precipitation sum, but such bias is below 15%. Further, relative higher skill of D3 for monsoon season is also evident from majority of error statistics either calculated on daily or monthly precipitation. Winter precipitation is highly overestimated (more than 150% bias) in all resolutions, however the overestimation is relatively lower during Pre-monsoon and might be partially explained with the under-
observation of solid precipitation by the tipping bucket gauges. Moreover, simulated precipitation variability clearly shows the ability of all WRF domains in broadly capturing the annual cycle of precipitation, featuring highest precipitation during the monsoon months, which is followed by an abrupt decrease in the post-monsoon season (Figure 3). D3 again shows a relatively higher skill in reproducing the annual cycle characteristics. For daily high-intensity precipitation, all domains overestimate winter and pre-monsoon (except D1) but
underestimate post-monsoon high-intensity precipitation (Figure 4). For dominant monsoon season, D1 underestimates, D2 overestimates but D3 satisfactorily reproduces daily high-intensity precipitation events.

As noted from the observed characteristics of diurnal cycle presented in Figure 5, all stations feature lull in morning precipitation. For the upper valley stations, distinct primary peaks are observed at mid-night while secondary peaks are observed in late afternoon to evening (1400 -1900 local hours (h)). In contrast, D2 and D3
feature a primary evening peak (1600 h) but a secondary peak at mid-night and overestimate the precipitation intensity from afternoon to late evening. The coarser D1 only reproduces the secondary afternoon peak and highly underestimates the night-time magnitude. D2 and D3 also underestimate the nocturnal precipitation intensity from 2200 - 0300 h and start precipitating 2-3 h earlier than observed. Contrary to upper valley stations, the ridge stations show a primary precipitation peak in the evening and a secondary peak at mid-night.
D1 only reproduces day time peak but start precipitating 2-3 h earlier. D2 and D3 reproduce the primary



evening peak timings albeit overestimating precipitation intensity from noon (1200 h) to late evening (2000 h), but the magnitude between mid-night and early-morning is well reproduced.

In contrast to upper valley and ridge stations that feature primary and secondary peaks, lower valley stations feature only peaks in the mid-night phase (Figure 5). Further, lower valley stations feature a linear precipitation increase from noon to mid-night and decrease from mid-night to morning (0900 h). D1 completely fails to capture the peak precipitation timings while D2 simulates observed mid-night peak shifted to the evening (05 h early). D3 well reproduces the midnight peak precipitation and its linear increase and decrease as observed, though slightly underestimating the precipitation intensity.

Overall, lower valley stations are characterized by a strong nocturnal precipitation maximum during monsoon season. The D1 simulation completely misses such pattern, as it could not adequately resolve the river valleys and ridges and the associated circulation characteristics due to coarse resolution. D2 largely captures the diurnal precipitation pattern but substantially overestimates precipitation intensities, whereas D3 largely improves the simulation of diurnal cycle and reduces the positive biases in precipitation intensities. Early precipitation triggering bias for day time peak in ridge and upper river valleys decreases from D1 to D2 with further improvement from D3. In terms of magnitude, all domains in general overestimate the evening peak but underestimate the nocturnal peak consistent with Collier and Immerzeel (2015) and Norris et al. (2016). Higher overestimation in D2 than in D3 may be attributed to explicitly resolved convection over coarse grid cells, indicating that convection permitting simulations on higher resolutions can further improve the model skill in reproducing the diurnal cycle characteristics, thereby eliminating the convection from large area and better representing the entrainment process. Although D3 shows best performance with closer agreement with observations and a better representation of the spatial and diurnal characteristics, there still is deficiency for the narrow and deep upper valleys like Rolwaling, where mountains and valleys are not fully resolved in the 30 arc second WRF topography.

**Temperature**

Observed temperatures range from -14 ℃ in the high-mountains in January to 21 ℃ in the lowlands during early monsoon (Figure 6a).

The error statistics calculated for elevation adjusted daily temperatures clearly show the best performance of D3 (followed by D2 and D1) in simulating the temperature, featuring low bias (warm or cold), smaller MAE, RMSE, RAR and relatively higher r against observations (Table 4). During monsoon, D2 features the same skill as D3. On utilizing monthly data for calculation, most error statistics show improvement in terms of biases. In general, all resolutions show a net cold biases in all seasons, except D3 in post-monsoon.

The observed seasonal cycle of temperature characterizing lowest temperatures in January and highest during monsoon months has been well reproduced in all WRF resolutions and is also in good accordance with previous studies from Rolwaling (Gerlitz et al. 2016; Müller et al. 2016b). The cold bias is found during winter and pre-monsoon season, which interestingly reduces with improving resolution from D1 to D3 (Figure 6a). The best result of the highest resolution can be attributed to the better representation of topography and associated local circulation characteristics, which develop due to radiative surface heating and cooling and moisture processes in complex mountain terrain. On the other hand, the high cold bias during winter and pre-monsoon may be linked to the overestimation of snowfall (precipitation) in high altitude areas, since overestimated snow increases the surface reflectance and the limited available net radiation is additionally utilized for snow melting (Immerzeel et



al., 2014). This in turn reduces the sensible heat flux and ultimately the temperature in high altitude regions. These cold bias patterns attributed to wet bias in precipitation with snow cover, moisture and evaporation feedbacks are consistent to those reported in mountainous region of greater Alpine (Haslinger et al., 2013).

The observed environmental lapse rates feature a bi-modal distribution with maxima in pre-monsoon and early post-monsoon and minima in early-winter and monsoon seasons (Figure 6b). Precisely, the observed lapse rate vary from -4.7 ºCkm-1 in November/December to above -6.3 ºCkm-1 in March/April with an annual average of -5.6 ºCkm-1, consistent with previous studies (Kattel et al., 2012; Salerno et al., 2015). All WRF resolutions are able to broadly reproduce the bi-modal distribution of lapse rates. Their magnitude during the monsoon months is relatively better reproduced, which is in agreements with results of Gerltiz et al., 2016, who investigated the seasonal cycle of the ERA-Interim internal lapse rate over the Rolwaling Himal. However, winter and spring season lapse rates are simulated higher than observed. Similar to other statistics, the skill of D3 is superior to D2 and D1 domains.

The distribution of temperature percentiles is remarkably reproduced by the WRF, irrespective of the domain resolution, particularly for the maximum temperatures (Figures 7). Lowest percentiles of daily minimum temperatures however are highly underestimated during winter, pre- and post-monsoon seasons due to cold biases simulated for such periods, with a relative improvement for high-resolution domains.

The observed annual averaged diurnal cycle features minimum temperature in early-morning (at 0600 h) and maximum temperature around noon which is earlier than the peak observed over low elevated regions in the South of the target area (Figure 8). The diurnal temperature variability is well reproduced by the WRF domains with a negligible delay. However, during the night and morning hours, simulated temperatures show a strong negative bias, particularly for D2 and D1, where the topography is significantly smoothed and topo-climatic processes are insufficiently captured. During afternoon and evening hours the negative bias is replaced by a slight warm bias in all domains. As discussed earlier for other temperature characteristics, the diurnal cycle are best reproduced in monsoon and post monsoon while winter and pre-monsoon features higher cold bias.

Another noteworthy feature in diurnal bias is the abrupt falling tendency towards higher magnitude negative bias between 0900 - 1000 h, which is more distinct in D2 and D3 (where valley and mountain are resolved) during post monsoon and monsoon. As majority of our stations are in river valleys or nearby slopes, we speculate 'perhaps' the physical mechanism that favoured for too much fog development (cooling and high moisture) in the model in those places might be responsible for this. This speculated environment for high fog type weather formation in model can cause low penetration of solar radiation into the ground as some of the available solar insolation in early morning hours is used to dissolve the unrealistic (may be not too widespread and deep) fog present in model (as described in Nasa Report, 2001).

**4.2 Seasonal and spatial distribution of surface variables**

Extensive validation of WRF simulated surface variables against station observations in terms of several measures suggests a satisfactory skill of higher resolution simulations, particularly of D3. Here, we report on the spatial distribution of simulated surface variables across different seasons. Since the skill of D2 is roughly comparable to that of D3, we further assess the spatial distribution of surface variables over the wider domain of D2.





Winter precipitation is associated with western disturbances, which mainly influence the western Himalayas but occasionally reach the study region, developing a northwest-southeast gradient. Such gradient is generally captured by all WRF resolutions (Figure 9). However, drier Rolwaling and Khumbu valleys and distinct precipitation gradients between valleys and tops are only captured by higher resolutions (D2 and D3) where D1

features a smoothed evenly distributed precipitation pattern. D3 and D2 simulate precipitation twice as much over the ridges as over the valleys. Additionally, although both D2 and D3 have similar spatial pattern, precipitation is more concentrated over mountain tops for D3 than for D2, indicating a more realistic altitudinal representation of spatial pattern. Nevertheless, winter precipitation is overestimated over the southern Himalayan slopes, the upper valleys and mountain tops in all WRF domains. Such overestimation mainly arises

from few events of westerly disturbances, whose intensities are overestimated in all WRF simulations as depicted in Figure 3. All WRF domains indicate a west-east gradient of winter precipitation, which is more clearly depicted from the larger D2 domain. Overall, the distribution pattern of winter precipitation and its overestimation is congruent with previous studies (Seko 1987; Putkonnen, 2004; Norris et al., 2016; Collier and Immerzeel, 2015).

Pre-monsoon precipitation over the study region is primarily generated by localized convective instability (due to local surface heating and uplifting) with moisture supply from local sources or by the occasional passage of westerly disturbances during early months. However, due to the location of the target area in Eastern Nepal, a substantial amount of precipitation occurs in late pre-monsoon season, when moisture supply from the Bay of Bengal starts to increase. Thus, the combination of all of these features results in an east-west and altitudinal

gradient of precipitation during the season. D1 underestimates pre-monsoon precipitation over Rolwaling, Khumbu and their adjacent southern areas, while it overestimates precipitation over the southern Himalayan slopes. D2 overestimates precipitation roughly over whole domain, and smoothly capture the mountain-top and valley-bottom contrast. This contrast (drier valleys but slightly moist slopes and ridges) is well reproduced by the D3 simulation only. From larger D2 domain, the pattern of east-west precipitation gradient is clearly

depicted (Figure S2, Supplement).

During monsoon season, micro/mesoscale convective processes take place due to intricate interactions of the synoptic weather systems with the complex topography, responsible for producing an extremely heterogeneous spatial distribution of precipitation (Seko, 1987; Norris et al., 2015). Such pattern is well simulated by all WRF resolutions (Figure 9) but other seasons alike D1 again simulate a smoothed pattern and substantially

underestimated precipitation. Unlike D1, higher resolution simulations are able to resolve the precipitation contrast between valleys and mountain slopes and ridges. For instance, low precipitation (<800 mm) at Manthali and Manebhanjyang stations in the lower river valley is completely absent in D1, while it is well reproduced by D2 and D3. Similarly, high precipitation pockets observed at Gumthang (northwest of D3) (Karki et al., 2017) as well as precipitation sums in the order of 2000 mm at stations Bhalukhop and Aiselukharka (at the mountain

slopes) are resolved in D2 and D3 only. Further, for Rolwaling Khola watershed and Khumbu region in the north of the high mountain ranges, the observed negative altitudinal gradient is evident in both simulations D3 and D2, while D1 fails to simulate this meso-scale feature. For example, observed precipitation at the lower valley station of Phakding in Khumbu region is in the order of 1000 mm, while it drops by one-third upstream at Phiriche and Pyramid stations. Likewise, in Rolwaling Khola watershed, precipitation at the basin outlet and

over the surrounding areas is around 2000 mm and sharply drops to one-fifth upstream.



Although the spatial patterns are broadly similar between D2 and D3, owing to more realistic representation of topography, the mountain valley precipitation contrast is better resolved in D3. Further, the reduced positive precipitation bias for lower river valleys in D3 compared to D2 is clearly discernable. Over the larger D2 domain, simulated precipitation features two maxima zones: 1) a narrow peak precipitation band along the first

mountain barrier, and 2) a relatively broader peak precipitation band at the Southern slopes of the Himalayan range (Supplement Figure 2). The river valleys between these two zones are found relatively drier. Such pattern is in agreement with the station observations (Karki et al., 2017), satellite-derived observed estimates (Bookhagen and Burbank, 2006; Shrestha et al., 2012) and simulations (Maussian et al., 2014; Gerlitz et al., 2015). Unlike D2, D1 fails to capture such spatial pattern, as topography, that plays important role in developing

localized convective activity and orographic precipitation, is not well represented at coarser grid scale.

Post-monsoon is the driest season over the study region, for which the spatial distribution is well reproduced by WRF, however with overall overestimation (Figure 9). Interestingly, larger aerial coverage map of D2 and D1 show a west-to-east gradient, which is in contrast to the observed east-to-west gradient (Supplementary Figure S2). This contrasting pattern may be linked to a rare heavy snowfall event in Annapurna region of Nepal

Himalaya between 13 and 15 October that resulted from the collision of westerly trough with the remnants of the tropical cyclone Hudhud formed over Bay of Bengal (Wang et al., 2015). As shown in Figure 10, individual station biases show a mixed pattern of wet and dry bias with improvement in D3 resolution compared to D2.

Summarizing, the better representation of the model-topography with increasing horizontal resolution better reproduces the interplay of the large scale circulation with the topography, which results in an improved

simulation of localized convective precipitation, orographic precipitation and thus in an amendment of the spatial precipitation pattern.

For temperature, the skill of WRF in simulating the spatial distribution is restricted by the degree of representation of the real topography by the considered resolutions, where coarser resolution shows a smoothed pattern, while finer resolution resolves the spatial variability (Figure 11). Winter temperatures vary from less

than -30 ºC at higher Himalayan tops to above +20 ºC in the southern river valleys. These pattern are realistic but limited observations (mostly in river valleys) do not represent the temperature at high mountains as resolved by WRF. The spatial distribution of temperature is broadly similar in all seasons with only change in range of temperature between high and low altitude stations. There is a cold bias in pre-monsoon and winter temperatures in all three domains that shows the largest magnitude for the coarse D1 domain. The improvement

(reduction of the cold bias) with respect to finer resolutions can be clearly seen from the Figure 6a. Fine resolutions better represent the topography and the local land surface features, allowing the model to better resolve the associated physical processes and their interaction with the atmospheric circulation. Overall the cold bias simulated for winter temperatures (Table 4) over the study area is generally dominated by the cold bias from the upper valley stations, as mostly lower valley stations feature a warm bias (Figure 12).


### 4.3 Altitudinal gradients of surface variables

The altitudinal distribution of spatially mean winter precipitation shows a peak of about 275mm at an elevation of 5000 masl in both D3 and D2 resolutions, while lower river valleys (~ 500 masl) experience lowest precipitation of 50 mm (Figure 13). In D1, the altitudinal dependency is completely missed for all seasons. The

altitude of simulated peak winter precipitation is in agreement with the observations and with other modelling



studies over Nepal and Himalayas (Seko, 1987; Putkonnen, 2004; Collier and Immerzeel, 2015; Norris et al., 2016).

In contrast to the winter precipitation peak at 5000 masl, the peak monsoonal precipitation of ~2700 mm is simulated around 3000 masl by D3 and at slightly lower elevations by D2. The precipitation sharply decreases both at higher and lower altitudinal belts. D1 fails to reproduce the characteristic precipitation gradient. Compared to D3, D2 overestimates precipitation below 3000 masl, which may be linked to the development of wider convective cells along the windward slopes of the mountain ranges. The altitudinal gradient of monsoonal precipitation is consistent with the observed gradient and the precipitation peak around 2500 masl over the whole Koshi basin (Salerno et al., 2015). Putkonen (2004) reported an elevation of peak monsoonal precipitation over Marsyangdi River Basin in western Nepal at 3000masl. Miehe (1990) detected a monsoonal precipitation peak between 3000 and 3600masl for the southern slopes of Jugal Himalaya, located in the west of our study area. Moreover, WRF (Collier and Immerzeel, 2015; Norris et al., 2016) or statistically based downscaled datasets (Gerlitz et al., 2015) also report similar altitudes of peak monsoonal precipitation over different regions of Nepal. Pre-monsoon and post-monsoon peak altitude is found in similar altitudinal range as during monsoon.

As discussed earlier, winter precipitation is affected by few extra-tropical cyclones (western disturbances), which are often formed in association with upper tropospheric Rossby waves. In these systems, warm and moist air masses are overlaid in a cold and dry air (frontal circulations). The uplift, which is necessary for the generation of precipitation, is dynamically caused by cross-barrier flows. In contrast, monsoonal systems, being convective in nature and originating from the surface, have lower vertical extent than extra-tropical systems (see variable PBL in Figure 2). Thus, the seasonal contrast in peak precipitation elevation between winter and summer is a direct result of the interplay of characteristic large scale circulation mechanisms and the local scale topography of the target region.

The altitudinal variation of temperature is found similar in all three resolutions with minor differences. The average lapse rates (all grid considered) calculated over D3 region for winter, pre-monsoon, monsoon and post-monsoon seasons are 7.4, 7.3, 5.6 and 6.8 °Ckm-1, respectively. Higher winter and pre-monsoon lapse rates from D3 are most probably affected by the cold bias observed over the upper valley regions, which is even higher for the coarser resolutions. The seasonal differences in lapse rates can be attributed to changes in humidity. The shallower lapse rate observed during monsoon results from the condensation in the middle atmosphere (with abundant moisture availability) that releases latent heat, thereby warming the higher mountain ranges (Kattel et al., 2012; Immerzeel et al., 2014). In addition, cloud cover can negatively influence the lapse rate (Kattel et al., 2014), since higher cloud coverage reduces the radiative energy causing shallower lapse rates.

### 4.4 Diurnal precipitation characteristics

Spatial maps of peak precipitation timing clearly show the capability of the D3 resolution in simulating the observed contrast in diurnal characteristics between the river valleys and the mountain slope/ridges (Figure 14). Though D3 accurately captures the mid-night peak evident in the deep and wider river valleys of Khumbu, it has still deficiencies in representing the nocturnal peak precipitation observed in the narrow river valley of Rolwaling. D2 reproduces the general diurnal contrast between river valleys and ridges, however the night time peak over the river valleys is shifted to late evening. In contrast, D1 fails to simulate any difference between river valleys and slopes/ridges, as noted earlier. Thee-hourly precipitation suggests that evening peak precipitation shifts more towards mid-night to morning hours from ridge to river valley locations in D3 (Figure



15). D2 triggers evening while D1 triggers afternoon peaks earlier. The underestimation of night time peak but overestimation of daytime peak magnitude is consistent to Norris et al. (2016), but our findings impressively suggest the capability of a kilometer scale simulation in better resolving the diurnal peak timings within the river valleys.

As described earlier in previous studies based on TRMM satellite estimated precipitation data (Bhatt and Nakamura, 2006; Sahany et al., 2010; Shrestha and Desar, 2014), mid night to early morning precipitation peak is observed over the southern foothills of Himalaya. This large scale feature is well reproduced by D2 while in D1 resolution, only evenly distributed afternoon to evening peaks are noticed (Figure 16).

Three-hourly precipitation over large D2 domain clearly shows the skill of convection permitting WRF
simulations in simulating the shift of afternoon/early-evening peak over high mountains to the midnight/early-morning over wide river valleys and foothills of Himalayas, consistent with earlier reports (Bhatt and Nakamura, 2006; Shrestha and Desar, 2014; Norris et al. 2016). D1 however triggers early convection (1200-1500 h) only. The shift in lull precipitation hours from morning towards day hours from north to south can also be clearly observed (Figure 17). These features are further supported by the plotted surface (10m) winds. From
0900 h until 1800 h, the surface wind plot shows the diversion of southeasterly warm monsoonal flow towards the Himalayan mountain range, which results in an orographic uplift with accompanied labilisation of the atmosphere. The model is capable in simulating both, the strong vertical ascent of air masses over the mountain ranges (see Figure 18) and the consequential monsoonal precipitation (Figure 17). After 1800 h, the flow over the mountain valleys reverses. The high mountain regions are dominated by a southerly flow, which follows the
direction of the major Himalayan valleys and might be interpreted as nocturnal drainage of cold air. The cold air converges with the monsoonal current over the Himalayan foothills, which, leads to the development of deep convection and the associated nocturnal precipitation peak.

Summarizing, D2 can resolve the large-scale night time peak as observed over the foothills, but features limited skill in fully reproducing the valley-based nocturnal peak in the study region. D3 substantially improves the
results, providing an almost realistic representation of the shape of the diurnal precipitation cycle and its peak timing within river valleys. Nevertheless, sub-kilometer scale simulation seems to be needed to resolve the topography, and thus the diurnal features in narrow river valleys.

**4.5 Mountain valley circulation**

Over the mountain tops and ridges, strong daytime heating in combination with moist monsoonal winds yield
strong convection and associated precipitation during the afternoon, while the valleys are characterized by dry conditions. On the other hand, nocturnal peak precipitation, evident first in the lower river valleys and then over the southern foot hills, is associated with the convergence of cold down-valley winds from radiatively cooled high mountains and the warm and moist monsoonal flow (Higuchi et al., 1982; Barros and Lang, 2003). Additionally, cold air pooling might cause a pronounced night time precipitation peak in lower river valleys
(Gerlitz et al., 2016). The three-hourly averaged vertical wind speed plotted along the cross-section AB (Figure 18) clearly demonstrates the skill of D3 and to-some-extent of D2 in reproducing the characteristics of mountain valley circulation, mainly owing to better representation of mountain slopes, ridges and valleys. In contrast, D1 fully fails to reproduce such characteristics, which is mainly responsible for its poor performance in reproducing the diurnal cycle. In D3 and D2, very strong convective activity (high positive vertical velocity as proxy) at the
mountain slopes and ridges occurs from late afternoon to evening hours. This convective activity weakens in the late evening and is replaced by subsidence during the night. The convergence of cold valley winds with the



monsoonal flow results in a night time precipitation peak over the river valleys. It is to note that D3 topography resolves many river valleys and ridges that are not evident in D2, thus the resultant steeper terrain yields more realistic strong and narrower (daytime) updrafts and (night time) downdrafts. As discussed in previous section, this improved representation of the mountain valley circulation produces realistic night time peaks over river

valleys such as Khumbu and lower reaches of Rolwaling in D3, which is not fully represent in D2. However, lack of strong night time convergence in overall leads to underestimated precipitation as noted by Norris et al. (2016). This may be due to: 1) too much release of moisture at the mountain slopes during the day (anomalously strong convection); 2) underestimation down valley winds due to smoothed topography.

Overall, it is found that adequate simulation of the diurnal cycle in the model is directly linked to adopting

higher resolutions, owing to their better representation of real topography and associated microphysical processes of the mountain-valley circulation, entrainment process and insolation.

### 5. Conclusions

This study uses one year of WRF simulation in three nested domains of 25 (non-convection permitting), 5 and 1 km (convection permitting) horizontal grid spacing over a highly structured Himalayan target area in order to

investigate the impact of model resolutions on reproducing the seasonal and diurnal characteristics as well as the spatial pattern of temperature and precipitation.

The overall evaluation of the WRF (using a network of high altitude and low altitude stations) clearly demonstrates the added value of high over coarse resolutions for both surface parameters, not only in terms of spatial pattern but also in terms of magnitude, seasonal cycle and diurnal characteristics. The better

representation of topographic features in high resolution WRF runs allows the realistic simulation of the mountain-valley circulation (convection) and other local scale processes and leads to an improved representation of the surface variables. The better simulation of monsoonal precipitation and temperature, which are strongly influenced by the mesoscale mountain valley circulation, further confirms this. Therefore, this assessment shows the potential of WRF high resolution (convection permitting) simulations for climate impact

studies, environmental applications and for investigations of the interplay of synoptic weather systems with local topography. The better simulation further shows the adequacy of WRF for future climate projection in the region. However, these conclusions are based on only one year of simulation and need to be further validated by means of longer term modelling studies. Further, for deep and narrow river valleys, sub-kilometer scale simulation with higher resolution of land use, land cover and topographic data are needed to fully resolve micro-

climatic features. It is worth to mention that (sub)-kilometer scale simulations for larger areas are computationally very expensive, however, our study indicates that slightly coarser (but still convection permitting) WRF simulations (with 5 km spatial resolution) capture the essential features of precipitation and temperature over the complex target area. A modelling chain, including dynamical downscaling of climate model results and subsequent statistical downscaling and terrain parameterization approaches (as proposed in

Gerlitz et al., 2015) to kilometer and sub-kilometer scale might be a feasible option for the region.  The main findings of this study are summarized follows:

- In overall day to day variation and hourly features of precipitation and temperature are better simulated by high resolution simulations (value added) with more realistic representation of magnitude during the dominating monsoon season than in other seasons.

- Precipitation in the 25 km simulation is underestimated and the spatial pattern is unrealistic, particularly the contrast between mountains and valleys is not adequately represented. Both 5 km and 1



km simulations broadly reproduce these features, but owing to better representation of topography in 1 km, the mountain valley precipitation contrast is better resolved in later. Further, the overestimation of precipitation in the 5 km simulation is highly reduced in the 1km run, with better representation of intense precipitation events.

- The altitudinal dependency of precipitation in high resolutions show peak precipitation at around 5000 masl during winter season and ~ 3000 masl during monsoon season, which is in agreement with observations. The low resolution domain completely fails to simulate the elevational gradient of precipitation.

- The representation of characteristic diurnal cycle of precipitation over the target region is impressively

well simulated by the high resolution domains, while the low resolution domain does not reproduce the nocturnal maxima over mountain valleys and the Himalayan foothills. The 5 km resolution domain shows a distinct nocturnal maximum of monsoonal precipitation over broad valleys and foothills only, which can be attributed to the convergence of down valley winds with the monsoonal flow. The 1 km domain substantially improves the results, providing an almost realistic representation of the shape of

the diurnal precipitation cycle and its peak timings in river valleys. Nevertheless, sub-kilometer scale simulation seems to be needed to resolve the topography and associated topo-climatic processes in very narrow river valleys.

**List of Figures and Tables:**

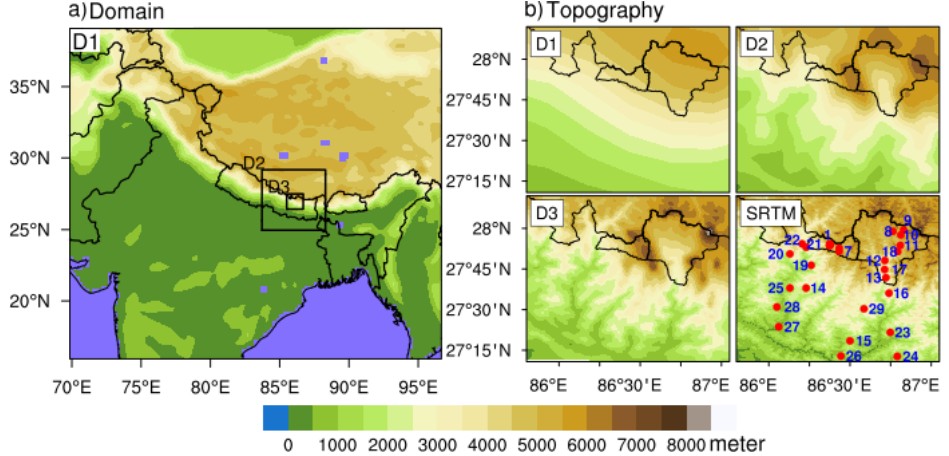

**Figure 1:** a) Domain boundary of three different WRF resolutions (left panel) with water area marked as blue, b) topography (right panel) of the study region in three different domain resolutions- 25km (D1), 5km (D2) and 1km (D3) - and real elevation from SRTM (30m resolution) with Rowaling (left basin) and Khumbu catchment (right basin) and, Nepal political

boundary (right panel) delineated. Meteorological stations (red marker) labeled as Table 2. But due to dense network in Rowaling only stations 1 and 7 out of 1 -7 are labeled.





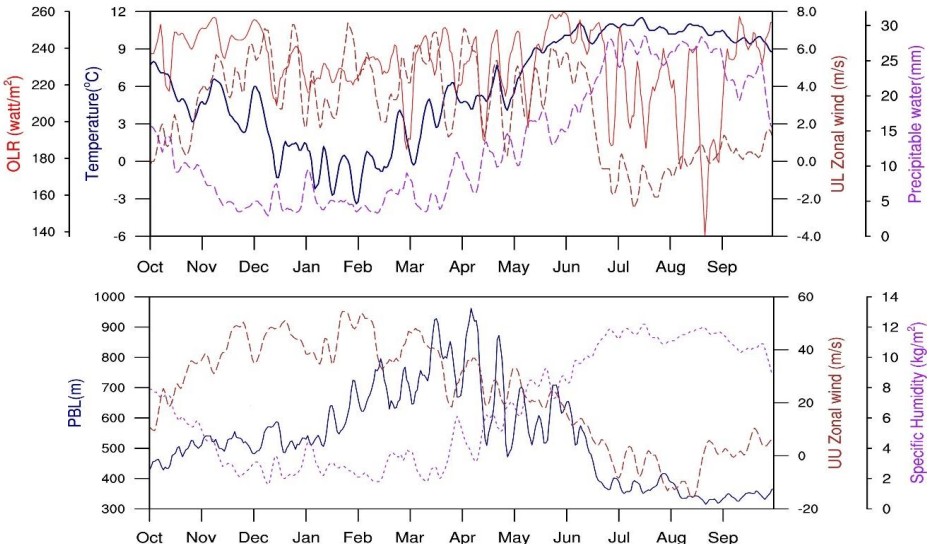

**Figure 2:** Time series of various parameters (D3 domain averaged); lower level (averaged for lowest 100 hPa) zonal wind (UL m/s) and air temperature (°C), perceptible water (mm) and, total out going long wave radiation (OLR) (watt/m²), upper level (300-150hPa averaged) zonal wind (UU), lower level (averaged for lowest 100hPa) specific humidity (g/kg), and planetary boundary layer (PBL) height (m) plotted as five days running mean.

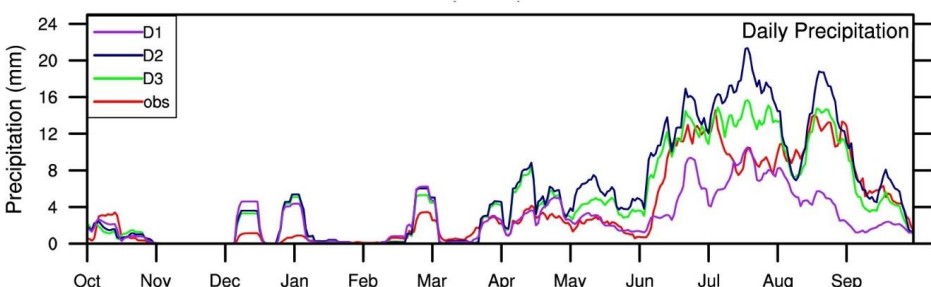

**Figure 3:** Daily station averaged precipitation (mm/day) from three WRF resolutions and observation (plotted as 10 days moving average). Observation abbreviated as obs hereinafter in all the figures herein after.

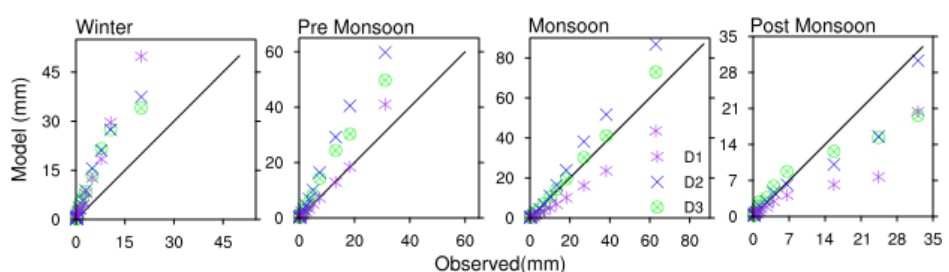

**Figure 4:** Percentile values comparison (1, 5, 10, 20, 30, 40, 50, 60, 70, 80, 90, 95, 99) for daily precipitation in different WRF resolutions with observation for four seasons. (Percentiles calculated for precipitation >=0.2mm/day therefore this has resulted in relatively higher value in other seasons too despite its lower total precipitation)





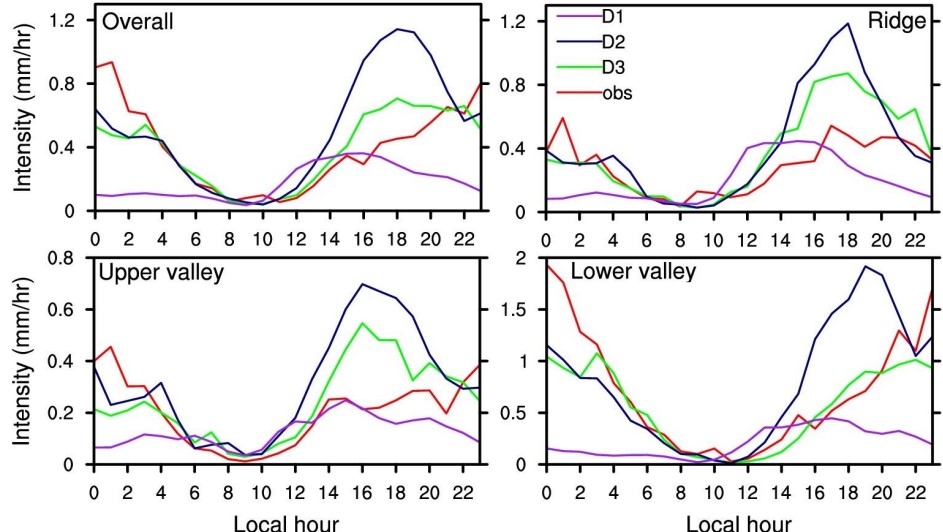

**Figure 5**: Diurnal precipitation during monsoon seasons in different WRF resolutions and observation, categorized into overall (all average), ridge, upper valley and lower valley.

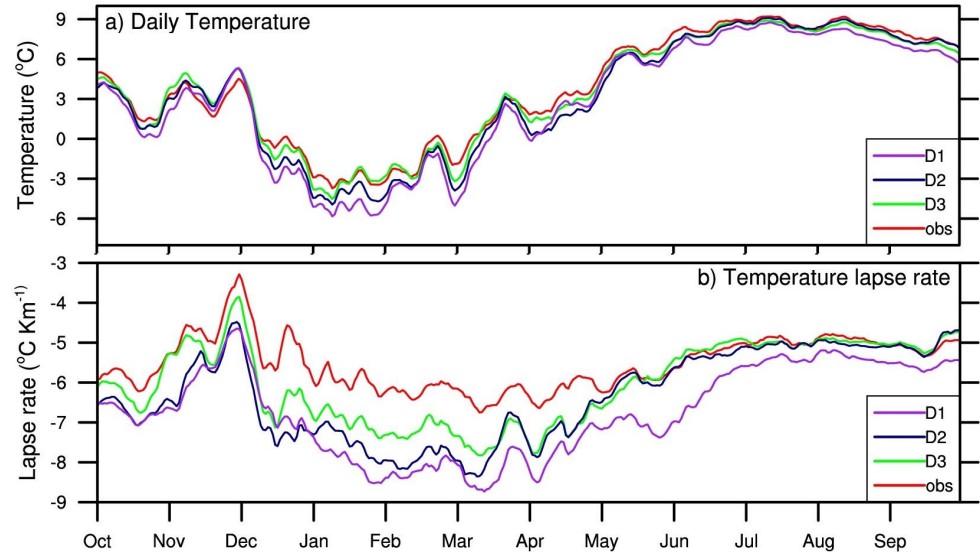

**Figure 6:** Daily station averaged a) temperature and b) its lapse rate in three WRF resolutions and observation (plotted as 10 days moving average).



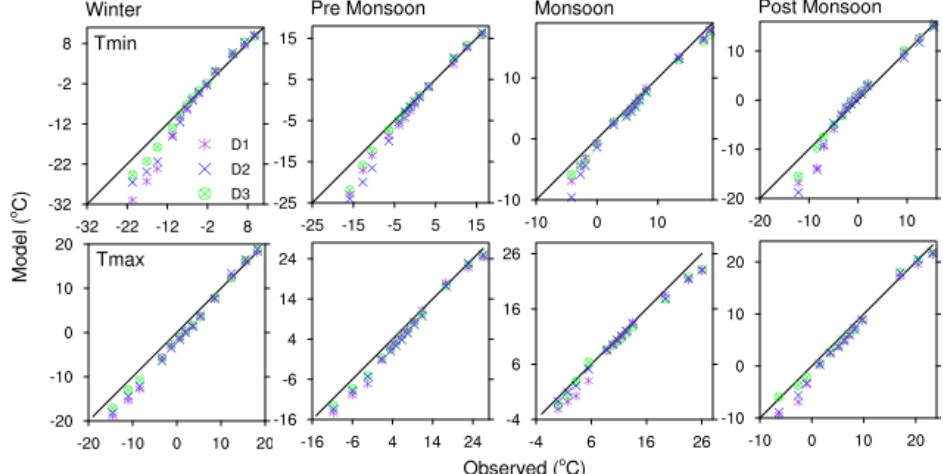

**Figure 7:** Percentile value comparison for daily minimum (Tmin upper panel) and maximum temperature (Tmax lower panel) (1, 5, 10, 20, 30, 40, 50, 60, 70, 80, 90, 95, 99 percentiles) in three WRF resolutions with observation for four different seasons.

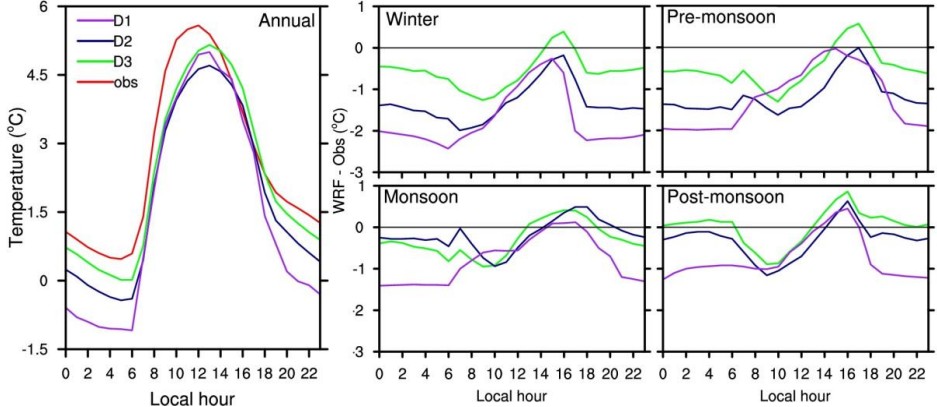

**Figure 8:** Diurnal mean annual temperature (left panel) and biases of temperature in all four seasons (right panel) in different WRF resolutions.



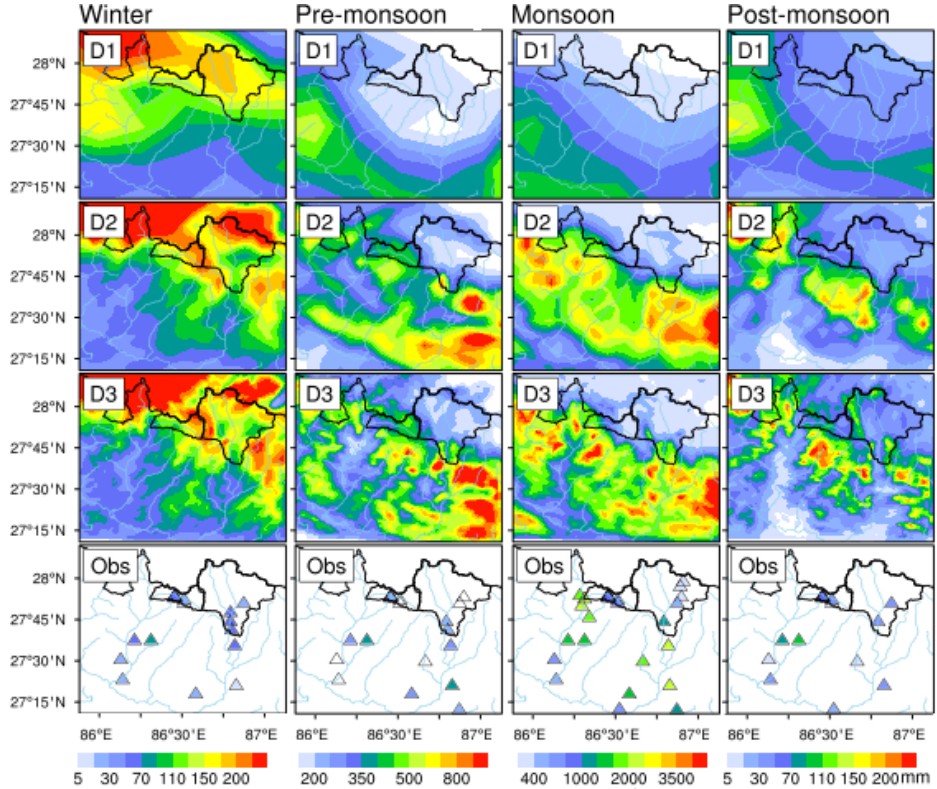

**Figure 9:** Spatial distribution of seasonal total precipitation in three different WRF resolutions (D3, D2 and D1) and observation with rivers over Nepal region delineated in light blue poly line, political boundary of Nepal and Khumbu, Rolwlaing catchment boundary in black.

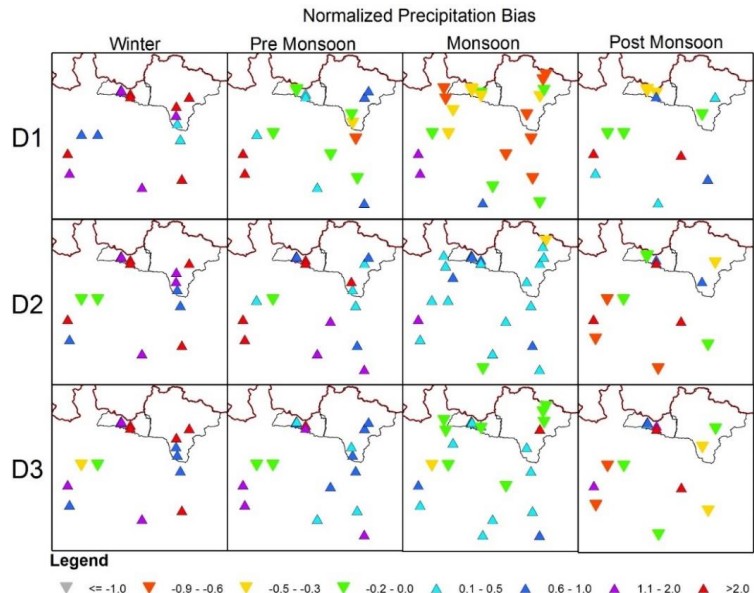




**Figure 10:** Station wise normalized bias [(WRF – Obs)/Obs] of precipitation for all WRF resolutions for all four seasons zoomed over station location only.

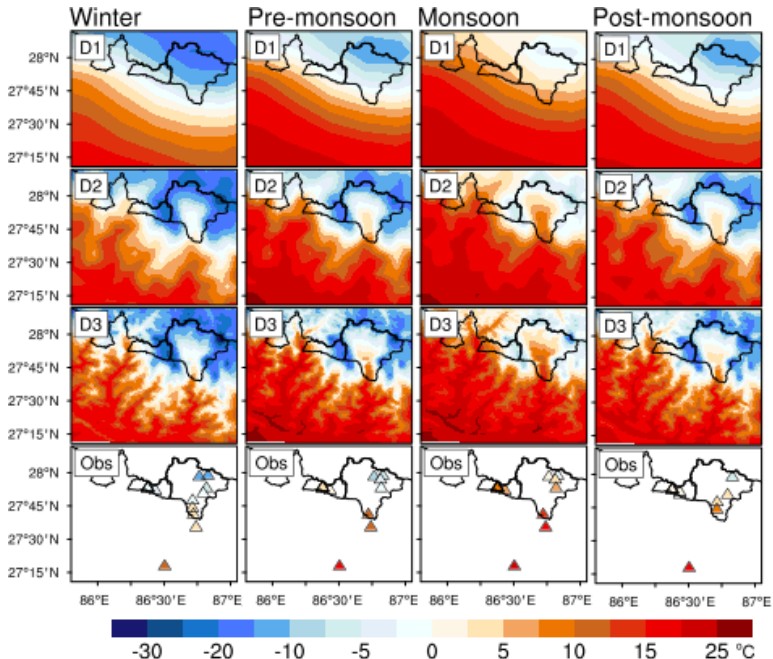

5      **Figure 11:** Spatial distribution of Temperature (°C) for four seasons in three WRF resolutions and observation.

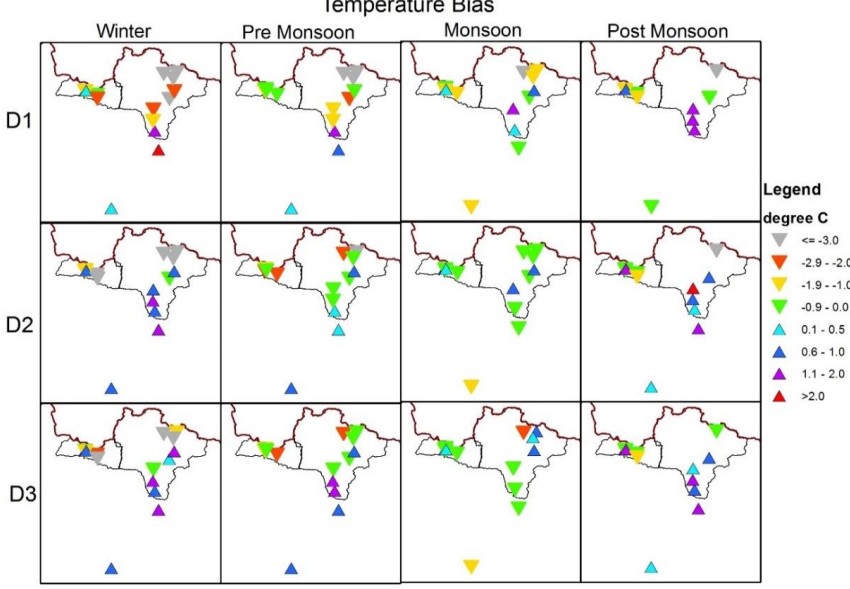

**Figure 12:** Station wise seasonal bias in temperature (°C) in all four seasons for three WRF resolutions zoomed over station locations only.



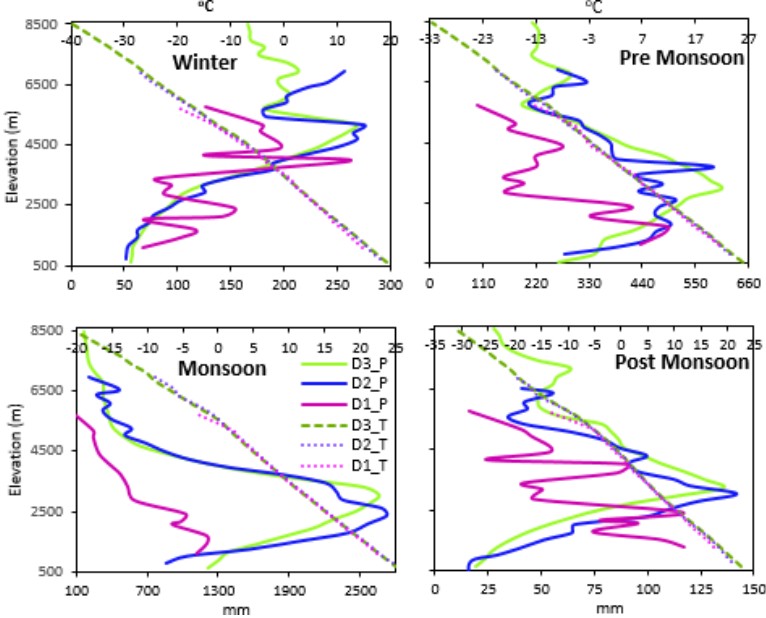

**Figure 13:** Altitudinal variation of precipitation and temperature in different WRF resolutions for all four seasons (over D3 area region only) [the abbreviation are D1, D2, D3 for three domains with P (precipitation) and T (temperature)]

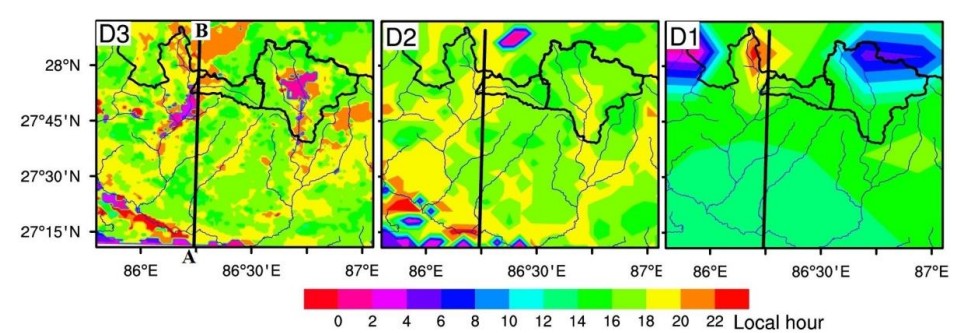

**Figure 14:** Peak precipitation hour (local hour) simulated by three WRF resolutions. Rivers over Nepal region are also delineated in polyline with political boundary of Nepal, and Rolwaling and Khumbu catchment boundaries in black. Section AB is also drawn.



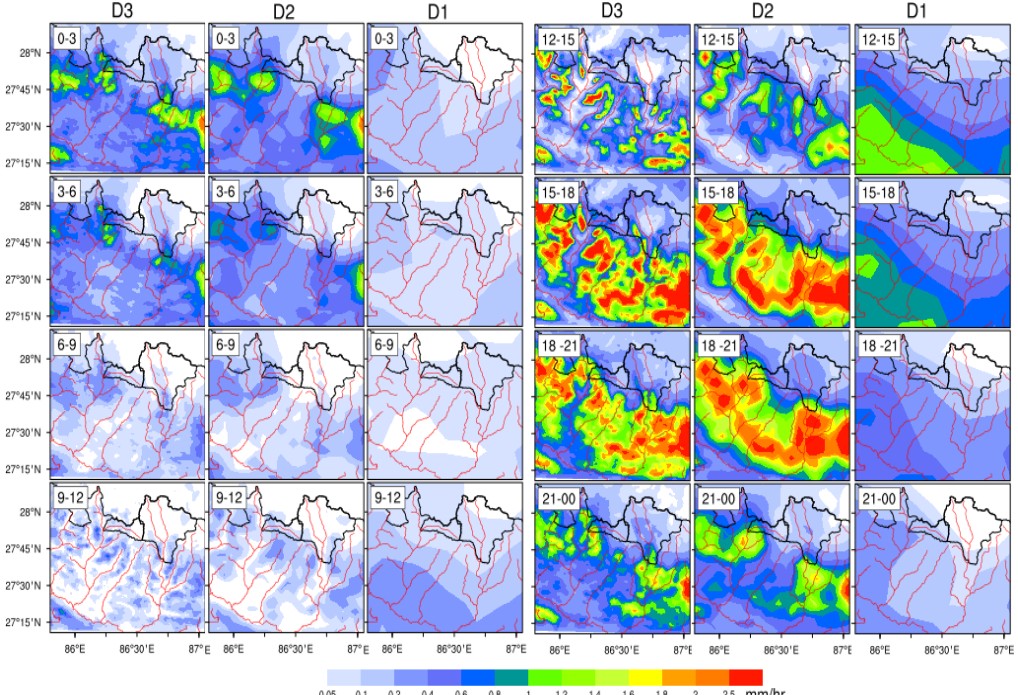

**Figure 15:** Spatial distribution of three hourly (local hour) average precipitation plot during monsoon season for three different WRF resolutions. Rivers over Nepal region are delineated in red lines and Rolwaling (left) and Khumbu(right) overlaid in black.

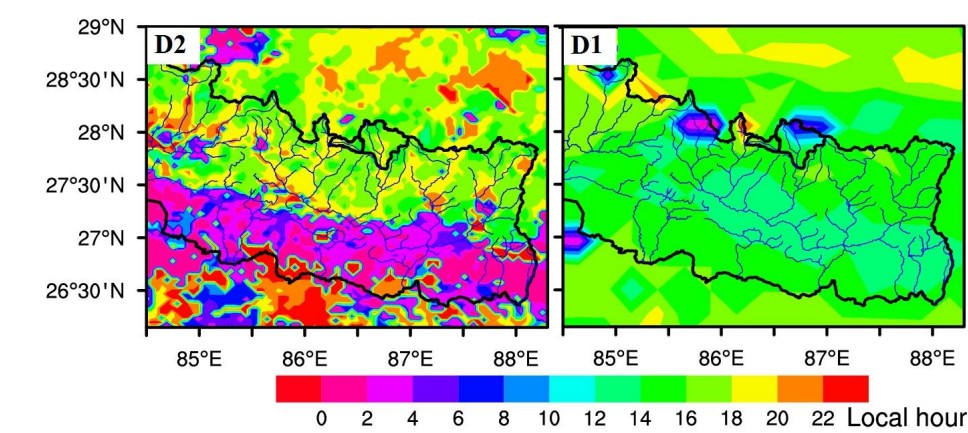

**Figure 16:** Spatial pattern of peak precipitation hour (local) during monsoon season in large area of D2 and D1.





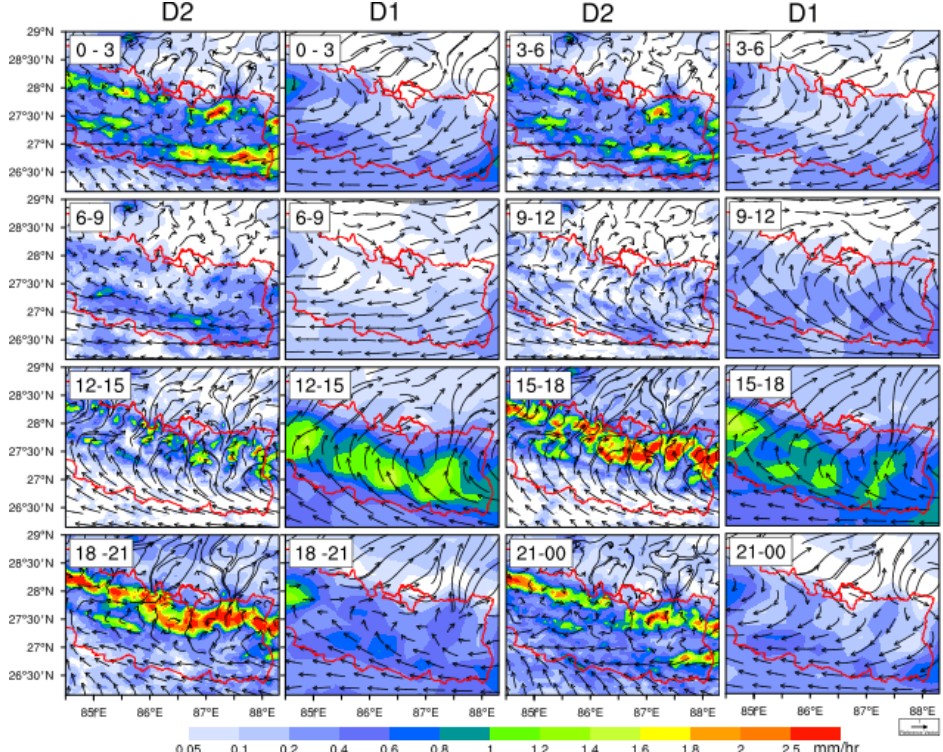

**Figure 17:** Spatial distribution of monsoon season three hourly (local hour) average precipitation in large area with surface wind (10m) vector overlaid, political boundary of Nepal in red with Khumbu and Rolwaling catchment in North.



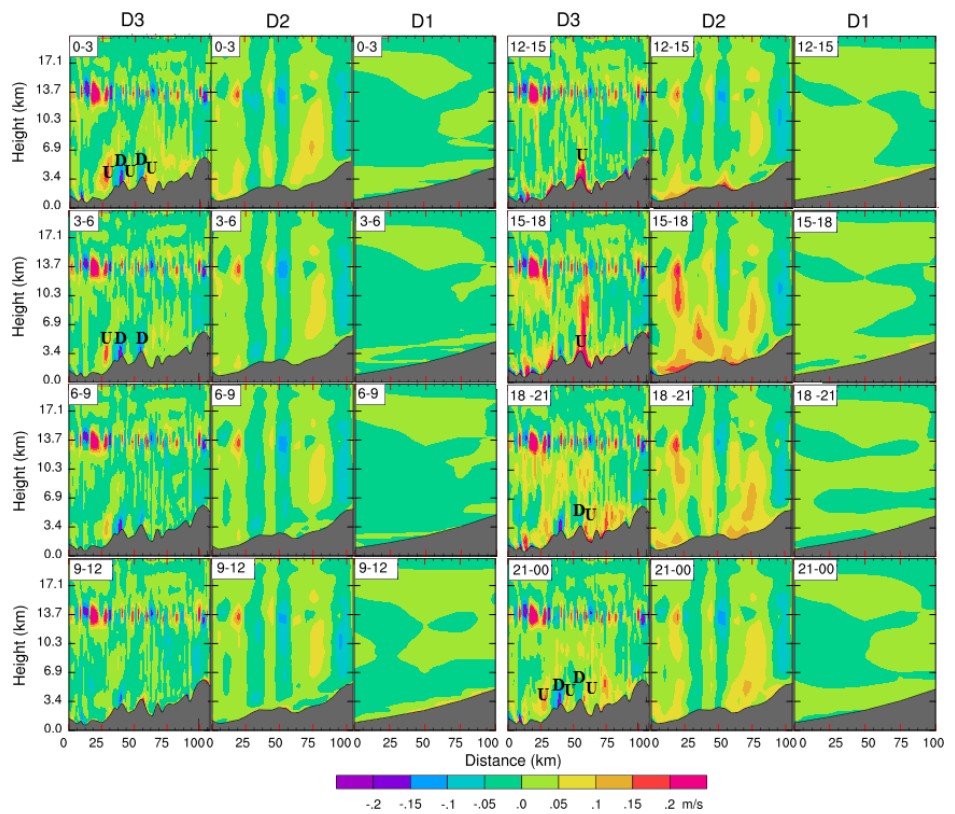

**Figure 18:** Three hourly (local hour) monsoon season average vertical velocity (m/s) in all three resolutions along the AB section with topography shaded in gray color. Few points of upward and downward flows are also marked as U and D, respectively.

## Tables

**Table 1:  Description of model configurations.**

| Description | Selection | References |
|---|---|---|
| Horizontal grid spacing | 25 km (D3), 5 km (D2), and 1 km (D1) | |
| Grid dimensions | 119 X 106,  101 X 96,  131 X 121 | |
| Vertical levels | 50 (11 lowest lying below 1km) | |
| Model top pressure | 50 hPa | |
| Nesting approach | One way | |
| Radiation | Community Atmosphere Model | Collins et al., 2004 |
| Microphysics | Morrison | Morrison et al., 2009 |
| Cumulus | Kain-Fritsch (except for 5 km and 1 km domains) | Kain, 2004 |
| Planetary boundary layer | MYNN level 2.5 | Nakanishi and Niino, 2006 |
| Atmospheric surface layer | Monin-Obukhov (revised MM5) | Jimenez et al., 2012 |
| Land surface | Noah-MP | Niu et al., 2011 |
| Forcing | ERA-Interim 0.75°×0.75°, 6 hourly | Dee et al., 2011 |



**Table 2: Summary of meteorological stations.**

(TP: Tipping Bucket, WG: Weighing Gauge, M: Manual, D: Disdrometer, T: Temperature, P: Precipitation, UH: University of Hamburg, DHM: Department of Hydrology and Meteorology, CNRS, IRD, Universite de Montpellier, T: Temperature, P: Precipitation)

| S.No. | Station Name | Latitude (Degree) | Longitude (Degree) | Elevation (masl) | Resolution | Parameters | Precipitation gauge | Data availability (%) T | P | Agency |
|---|---|---|---|---|---|---|---|---|---|---|
| 1 | Gompa | 27.905 | 86.375 | 3908 | Hourly | T, P | TP | 52 | 52 | |
| 2 | T1oben | 27.897 | 86.374 | 4055 | Hourly | T, P | TP | 100 | 100 | |
| 3 | T1unten | 27.901 | 86.376 | 3739 | Hourly | T, P | TP | 99 | 100 | |
| 4 | T2unten | 27.899 | 86.379 | 3750 | Hourly | T, P | TP | 99 | 100 | UH |
| 5 | T2oben | 27.893 | 86.376 | 4170 | Hourly | T, P | TP | 100 | 100 | |
| 6 | Na | 27.878 | 86.434 | 4219 | Hourly | T, P | TP | 100 | 100 | |
| 7 | Yalun | 27.859 | 86.434 | 5032 | Hourly | T, P | TP | 28 | 100 | |
| 8 | Chungrinup | 27.982 | 86.765 | 5700 | Hourly | T | | 86 | | |
| 9 | Kalapathar | 27.990 | 86.830 | 5600 | Hourly | T, P | TP | 76 | 74 | EVK2CNR |
| 10 | Pyramid | 27.959 | 86.813 | 5050 | Hourly | T, P | TP | 50 | 55 | |
| 11 | Phiriche | 27.895 | 86.819 | 4260 | Hourly | T, P | TP | 73 | 71 | |
| 12 | Namche | 27.802 | 86.714 | 3570 | Hourly | T, P | TP | 56 | 56 | |
| 13 | Lukla | 27.697 | 86.721 | 2660 | Hourly | T, P | WG | 88 | 63 | DHM |
| 14 | Jiri | 27.633 | 86.233 | 2003 | Daily | P | M | | 92 | |
| 15 | Okhaldhunga | 27.308 | 86.504 | 1725 | Hourly | T, P | D, M | 100 | 99 | |
| 16 | Bhallukhop | 27.601 | 86.740 | 2575 | Daily | T, P | TP | 77 | 86 | CNRS, IRD |
| 17 | Phakding | 27.747 | 86.713 | 2619 | Daily | T, P | TP | 50 | 100 | |
| 18 | Pangboche | 27.857 | 86.794 | 3950 | Daily | T, P | TP | 100 | 100 | |
| 19 | Chankhu | 27.773 | 86.262 | 1397 | Hourly | P | TP | | 37 | |
| 20 | Gongar | 27.843 | 86.216 | 1343 | Hourly | P | TP | | 31 | |
| 21 | Lamabagar | 27.905 | 86.205 | 1987 | Hourly | P | TP | | 50 | |
| 22 | Rikhu | 27.884 | 86.234 | 2088 | Hourly | P | TP | | 33 | |
| 23 | Aisealukhark | 27.360 | 86.749 | 2063 | Daily | P | M | | 100 | DHM |
| 24 | Diktel | 27.213 | 86.792 | 1613 | Daily | P | M | | 75 | |
| 25 | Kabre | 27.633 | 86.133 | 1755 | Daily | P | M | | 100 | |
| 26 | ManeBhanjyang | 27.215 | 86.444 | 1528 | Daily | P | M | | 67 | |
| 27 | Manthali | 27.395 | 86.061 | 499 | Daily | P | M | | 100 | |
| 28 | Melung | 27.517 | 86.050 | 1536 | Daily | P | M | | 92 | |
| 29 | Salleri | 27.505 | 86.586 | 2384 | Daily | P | M | | 92 | |

**Table 3: Seasonal precipitation error statistics (pooled data) based on monthly and daily precipitation.** * D1, D2, D3 is for three different domains, NBIAS: Normalized bias %, MAPE: Mean absolute percentage error r: correlation, RMSE: root mean squared error, RAR: RMSE/standard deviation of observed data.

| Error Statistics | Based on Monthly Precipitation | | | | Unit | Based on Daily Precipitation | | | |
|---|---|---|---|---|---|---|---|---|---|
| | Winter | Pre-Monsoon | Monsoon | Post-Monsoon | | Winter | Pre-Monsoon | Monsoon | Post-Monsoon |
| NBIAS D3 | 144.4 | 77.8 | 13.8 | 5.5 | % | 150 | 78.1 | 14.5 | 15.5 |
| NBIAS D2 | 165.0 | 109.6 | 36.9 | -0.4 | % | 172 | 107.1 | 35.9 | 8.0 |
| NBIAS D1 | 189.5 | 18.9 | -44.5 | 10.0 | % | 184 | 17.1 | -42.6 | 9.4 |
| MAPE D3 | 176.4 | 104.6 | 63.4 | 118.6 | % | 322 | 205.5 | 111.0 | 182.0 |
| MAPE D2 | 197.8 | 132.8 | 81.5 | 113.7 | % | 340 | 234.3 | 131.6 | 171.7 |
| MAPE D1 | 198.4 | 72.6 | 58.7 | 119.5 | % | 350 | 145.0 | 83.6 | 157.3 |
| RMSE D3 | 32.2 | 81.9 | 146.0 | 22.1 | mm | 4 | 8.7 | 16.6 | 3.9 |
| RMSE D2 | 36.0 | 106.2 | 191.8 | 23.6 | mm | 4 | 10.4 | 19.5 | 4.1 |
| RMSE D1 | 33.3 | 64.0 | 212.7 | 24.5 | mm | 5 | 6.6 | 13.9 | 3.4 |
| RAR D3 | 3.9 | 2.2 | 0.8 | 0.8 | | 2 | 1.7 | 1.3 | 1.1 |
| RAR D2 | 4.3 | 2.8 | 1.1 | 0.9 | | 2 | 2.0 | 1.5 | 1.2 |
| RAR D1 | 4.0 | 1.7 | 1.2 | 0.9 | | 3 | 1.3 | 1.1 | 1.0 |
| r D3 | -0.07 | 0.46 | 0.70 | 0.70 | | 0.32 | 0.33 | 0.29 | 0.25 |
| r D2 | -0.01 | 0.40 | 0.80 | 0.60 | | 0.31 | 0.28 | 0.25 | 0.18 |
| r D1 | 0.00 | 0.22 | 0.40 | 0.60 | | 0.32 | 0.27 | 0.25 | 0.42 |

**Table 4. Seasonal error statistics (pooled data) based on monthly and daily average temperature.**

(abbreviation same as Table 3 for common statistics between two tables additionally Bias: WRF – Obs, MAE: Mean absolute error)

| Error Statistics | Based on Monthly average temperature | | | | | Based on Daily average temperature | | | |
|---|---|---|---|---|---|---|---|---|---|
| | Winter | Pre-Monsoon | Monsoon | Post-Monsoon | Unit | Winter | Pre-Monsoon | Monsoon | Post-Monsoon |
| BIAS D3 | -0.36 | -0.34 | -0.33 | 0.31 | ºC | -0.20 | -0.38 | -0.33 | 0.25 |
| BIAS D2 | -1.16 | -1.10 | -0.22 | -0.10 | ºC | -0.93 | -1.03 | -0.23 | -0.07 |
| BIAS D1 | -1.94 | -1.44 | -0.78 | -0.45 | ºC | -1.58 | -1.18 | -0.78 | -0.48 |
| MAE D3 | 1.50 | 1.09 | 0.62 | 1.04 | ºC | 1.63 | 1.36 | 0.75 | 1.32 |
| MAE D2 | 2.03 | 1.49 | 0.58 | 1.33 | ºC | 2.19 | 1.75 | 0.74 | 1.60 |
| MAE D1 | 2.55 | 1.83 | 1.02 | 1.64 | ºC | 2.72 | 1.90 | 1.16 | 1.68 |
| RMSE D3 | 1.82 | 1.34 | 0.84 | 1.31 | ºC | 2.08 | 1.73 | 1.03 | 1.70 |
| RMSE D2 | 2.75 | 1.88 | 0.72 | 1.90 | ºC | 2.93 | 2.23 | 1.02 | 2.14 |
| RMSE D1 | 3.56 | 2.52 | 1.40 | 2.28 | ºC | 3.69 | 2.70 | 1.59 | 2.29 |
| RAR D3 | 0.29 | 0.19 | 0.15 | 0.25 | | 0.32 | 0.24 | 0.18 | 0.31 |
| RAR D2 | 0.44 | 0.27 | 0.13 | 0.36 | | 0.44 | 0.32 | 0.18 | 0.39 |
| RAR D1 | 0.57 | 0.36 | 0.25 | 0.43 | | 0.56 | 0.38 | 0.28 | 0.41 |
| r D3 | 0.98 | 0.99 | 0.99 | 0.97 | | 0.97 | 0.98 | 0.99 | 0.96 |
| r D2 | 0.98 | 0.99 | 0.99 | 0.96 | | 0.96 | 0.98 | 0.98 | 0.95 |
| r D1 | 0.97 | 0.99 | 0.99 | 0.95 | | 0.95 | 0.98 | 0.98 | 0.95 |

**Competing interests**

The authors declare that they have no conflict of interest.

**Acknowledgements**

The ERA-Interim data set are freely available from ECMWF. The authors would like to thank the Department of Hydrology and Meteorology (Nepal), EvK2CNR (Italy), TREELINE project from Institute of Geography, University of Hamburg, and Chair of Soil Science and Geomorphology, University of Tübingen (Germany) and Laboratoire Hydrosciences (CNRS, IRD, Université de Montpellier, France) for providing the meteorological data. Specially, Johannes Weidinger, Dr. Pierre Chevallier and Gian Pietro Verza support during data acquisition and Dr. Lindsey Nicholson, Dr. Emiley Collier, and Eleonore Schenk support and suggestions in data interpretation and model setup is highly acknowledged. Further, we are grateful to Dr. Meghanath Dhimal and, Prof. Dr. Bodo Ahrens and his team's for their suggestions and inputs on preliminary results. Ramchandra Karki's PhD scholarship was supported by Deutscher Akademischer Austauschdienst (DAAD) under the Research Grants—Doctoral Programmes in Germany, through University of Hamburg, Germany. Further, we acknowledge the TREELINE project funded by the German Research Foundation (SCHI 436/14-1, BO 1333/4-1, SCHO 739/14-1).

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
