# Peer review of "Quantifying the added value of convection-permitting climate simulations in complex terrain: A systematic evaluation of WRF over the Himalaya"

_Earth System Dynamics, 2017_

## Referee Comment (RC1) · Anonymous Referee #1 · 3 May 2017

General Comments: In this manuscript, the authors study the 'Added value (skill) of high resolution climate models (WRF simulations) in simulating the rainfall and temperature over Himalayan region'. Study of model resolution over complex hilly terrain is indeed important especially over the Himalayan mountain ranges where the rainfall is largely influenced by both the local factors as well as large scale circulations. In addition to that the paper also explains some of the important feature of precipitation over study region such as diurnal variations and spatial variability.

Specific Comments: The authors have selected a year of data with two initialization

conditions and compared the results of different horizontal resolution across different seasons viz. winter, pre-monsoon, monsoon, and post-monsoon. The skill of model resolution (25 km, 5km, and 1 km) is compared with observational station dataset at different altitudinal ranges. The description of the model setup and configuration part is nicely written in the manuscript. Although the manuscript is well written and results are properly explained for most of the cases, there are few areas where the explanation is inadequate (see the discussion in the comment section), nevertheless they are acceptable if revised. Therefore, I am suggesting a minor revision of this manuscript. I have some very minor comments provided below.

Minor comments: 1. The biases or spurious influence of the boundaries in the regional climate model can be reduced by nudging. Here authors can provide little more information about the details of the nudging.

2. In figure 3 'Daily station averaged precipitation' during mid-monsoon season (July-August) is not very well represented by D3, when compared with the observations. Authors need to comment on this finding. Again it will be really interesting to know why authors chose to use 10-day moving average.

3. In figure 5 'Diurnal precipitation during monsoon seasons' across D2 and D3 are close to observed in lower valley regions. However, for the remaining cases except for the morning precipitation, the precipitation is either overestimating (after noon) or underestimating (before morning). Authors can provide a little explanation on this.

4. In the explanation of figure 9 the authors have commented on west-east gradient during winter precipitation. However, I do not find a clear west-east precipitation gradient in all the three domains as well as in the observations. In fact I can see a north-south gradient. Authors needs to explain this with more clarity.

5.Figure 9: The difference between D2 and D3 is less for the winter season, pre-monsoon season and the monsoon precipitation.

[Figure]

6. Figure 10: For monsoon season the bias is more over southern region which is comparatively low elevation region... please provide a comment on this?

7. The authors argued that the pre-monsoon precipitation over the study region is mostly due the local scale circulations from local moisture source. If possible please provide an explanation with figure.

8. I see most of the local scale features in D2 and D3 are closely resemble with each other. How good is D3 compared to D2? The authors can briefly state this in the conclusions.

9. Although, it is not in scope of this manuscript, the authors can briefly comment on interaction between the westerlies and monsoonal circulation.

---

## Referee Comment (RC2) · Anonymous Referee #2 · 9 May 2017

The manuscript's goal to better understand the importance and effects of grid resolution on climate simulation in complex terrain is important. Especially, there is still a lack of studies with regional climate models in the convection permitting scale range. The manuscript is well designed and written in general too. I have two major issues (or better to say: wishes) and a few minor comments as given below. Therefore, I suggest a major revision.

The manuscript discusses simulations with 25, 5, and 1 km. One result is that the 25-km simulation misses the observed diurnal cycle and the authors explain this by a

lack of ability of the simulation to resolve the complex orography. Another explanation might be that convection parameterisations have an issue in simulating a proper diurnal cycle. It would add value to the manuscript if the authors can do simulations with 5 or 1 km set-ups, but the orography smoothed to the 25-km simulation orography. This would strongly support the authors conclusion.

There is no shallow convection parameterisation applied in the 5-km set-up? Please, clarify. I assume simulations with or without shallow convection differ substantially in the case of the 5-km set-up.

Minor comments:

1. The first part of the title is too general and in the second part the reference of the comparison is missing (perhaps better to replace comparison with evaluation). I suggest the title to be revised.

2. Please, give a reference to the TREELINE project already in the introduction.

3. Sec. 2: decrease of the precipitation in the valleys with and because of north-south orientation?

4. Eq. 1 & 2: Index m missing at the Os in the numerators?

5. You refer to Fig. S1 two times. If it is an important Fig. think about putting it in the main text.

6. Page 9, line 25: These are the mean observed temperatures probably?

7. At the end of Sec. 4.1 there is some speculation which should be avoided.

8. Page 12, line 23: Earlier in the manuscript it was mentioned that bias correction was applied for temperature. Why does the coarse simulation show smoothness after bias correction using the fine-scale orography?

9. Fig. 2 can be omitted. I think the text can be shortened quite a bit if carefully

reconsidered. There are some minor language issues like missing "a"s, "s"s and "the"s.

---

## Author Comment (AC1) · 23 May 2017

**Reviewer 1:**

We sincerely thank the reviewer for taking the time to thorough review our manuscript and for providing the helpful and constructive comments and suggestions for improvement. We hope that we have addressed the Referee's comments. Our responses are in black color font.

*General Comments: In this manuscript, the authors study the 'Added value (skill) of high resolution climate models (WRF simulations) in simulating the rainfall and temperature over Himalayan region'. Study of model resolution over complex hilly terrain is indeed important especially over the Himalayan mountain ranges where the rainfall is largely influenced by both the local factors as well as large scale circulations. In addition to that the paper also explains some of the important feature of precipitation over study region such as diurnal variations and spatial variability.*

*Specific Comments: The authors have selected a year of data with two initialization and compared the results of different horizontal resolution across different seasons viz. winter, pre-monsoon, monsoon, and post-monsoon. The skill of model resolution (25 km, 5km, and 1 km) is compared with observational station dataset at different altitudinal ranges. The description of the model setup and configuration part is nicely written in the manuscript. Although the manuscript is well written and results are properly explained for most of the cases, there are few areas where the explanation is inadequate (see the discussion in the comment section), nevertheless they are acceptable if revised. Therefore, I am suggesting a minor revision of this manuscript. I have some very minor comments provided below.*

Minor comments:

*1. The biases or spurious influence of the boundaries in the regional climate model can be reduced by nudging. Here authors can provide little more information about the details of the nudging.*

Response: We have rephrased and added following text in the revised manuscript (line 21 to 27 page 5)

> Mesoscale climate models may have difficulties in representing the large-scale features (Jones et al., 1995), therefore nudging is often used for longer simulations to prevent the downscaling model from simulating completely different and unrealistic drift from driving model, and to ensure the timings of synoptic disturbances to be in phase with driving model (Von Storch et al., 2000, Pohl and Crétat, 2014). But, the value added by nudging depends on skill of forcing data. Two different types of nudging termed spectral (wave numbers to filter large and small scale features) and grid analysis (in every grid cell) exist. Over Himalaya, the sensitivity study of both types of the techniques by Norris et al. (2016) found no significant quantitative differences in the precipitation distribution. Thus, we have applied grid analysis nudging only to D1 and merely for the horizontal winds, potential temperature, and water vapor mixing ratio in the vertical levels above the planetary boundry layer (PBL) and above the lowest 15 model levels thereby allowing meso-scale forcing in the PBL as the strong nudging even in the PBL where most of the atmospheric phenomena takes place, can prevent the downscaling model from representing meso-scale (small scale features) processes (Alexandru et al., 2008). This strategy is based on previous studies (Otte et al., 2012; Collier and Immerzeel, 2015), which have demonstrated the improved simulation of mean and extremes of surface variables with the application of such mild nudging.

*2. In figure 3 'Daily station averaged precipitation' during mid-monsoon season (JulyAugust) is not very well represented by D3, when compared with the observations. Authors need to comment on this finding. Again it will be really interesting to know why authors chose to use 10-day moving average.*

Response: We had missed to mention this result, therefore, we have added following lines in the revised manuscript (line 28 page 8) now;

> However, precipitation is consistently overestimated by both high resolutions during the whole two weeks period in mid-monsoon (Jul –Aug), likewise the dry bias in D1 is significantly reduced with its pattern more closer to observation as compared to other monsoon days. Similar overestimation in WRF simulations identified by Raju et al. (2014) is attributed to excessive moisture transport from Bay of Bengal and western pacific towards the monsoon dominated regions due to the strong low level easterly wind bias. Hence, we speculate the similar mechanisms to be responsible for our simulations too.

- As the day to day fluctuation in precipitation is high, we smoothed using 10 days moving average to make the comparison easier to follow.

*3. In figure 5 'Diurnal precipitation during monsoon seasons' across D2 and D3 are close to observed in lower valley regions. However, for the remaining cases except for the morning precipitation, the precipitation is either overestimating (after noon) or underestimating (before morning). Authors can provide a little explanation on this.*

Response: We agree with the reviewer that particularly in the upper valley stations (with dominancy of Rolwaling stations) where narrow river valley is not resolved even in 1km topography, the pattern differs from observation. But, ridge stations captures the pattern though the magnitude of day time precipitation is overestimated.

Hence, we have rewritten it as (line 22 page 9) ;

> Although D3 shows best performance with closer agreement with observations and a better representation of the spatial and diurnal characteristics, there still is deficiency for the narrow and deep upper valleys like Rolwaling- **reflected with relatively poor performance in upper valley stations -** where mountains and valleys are not fully resolved in the 30 arc second WRF topography.

- In general, overestimated day time peak and underestimated night time peak magnitude are common deficiencies in WRF simulation. The possible reasons for this were discussed in our manuscript line 7 page 15 as
  This may be due to: 1) too much release of moisture at the mountain slopes during the day (anomalously strong convection); 2) underestimation of down valley winds due to smoothed topography.

*4. In the explanation of figure 9 the authors have commented on west-east gradient during winter precipitation. However, I do not find a clear west-east precipitation gradient in all the three domains as well as in the observations. In fact I can see a north-south gradient. Authors needs to explain this with more clarity.*

Response: Corrected as north south gradient for the Figure 9 case which covers relatively smaller area. Nevertheless, if we see in original supplement Figure 2 for larger area, the gradient from west-east is obvious due to weakening of westerlies towards east (also mentioned in the manuscript).

*5. Figure 9: The difference between D2 and D3 is less for the winter season, premonsoon season and the monsoon precipitation.*

Response: We agree with the reviewer that essential features are the same between D2 and D3, but there is relative improvement in later. We have also mentioned that in our conclusions, thus the choice of high resolutions depends on available computing resources and time.
When we look it in detail in Figure 9 then the better resolving of topo-climates in D3 (mountain, valley and ridge precipitation contrast) owing to better representation of topography is evident.
For instance, if we follow the precipitation pattern near the outlets of the basins and river valleys between D2 and D3, the relative reduction in overestimated magnitude of precipitation in later is apparent for all of these seasons. Further, the pattern closely follows more realistic topographic features at D3. The overall improvement in D3 can be also seen in error statistics in Table 3.

*6. Figure 10: For monsoon season the bias is more over southern region which is comparatively low elevation region... please provide a comment on this?*

Response: These texts are added in the revised manuscript (line 34 page 12.

> In contrast to other seasons, the warm biases evident in lower river valley station locations reverses to cold during monsoon season in high resolutions, which can potentially be associated with wet bias in precipitation in those locations with moisture and evaporation feedbacks as discussed earlier.
> (* our original Figure 10 supports this attribution)

*7. The authors argued that the pre-monsoon precipitation over the study region is mostly due the local scale circulations from local moisture source. If possible please provide an explanation with figure.*

Response: We rephrased (add some texts too) and added reference to the previous study which had analyzed this;

> In study area section (line 19 page 4):
> The season is dominated with strong dry northwesterly winds. However, despite unfavourable synoptic environment, precipitation usually observed in the evening is generated primarily by localized convective instability with heating and is supported by the mountain uplift in the areas of surplus moisture (Shrestha et al., 2012)
> In result section (line 15 page 11):
> We had explained, and indicated not only local but there is also some contribution of synoptic scale systems in the manuscript. Now reference will be added.

> Pre-monsoon precipitation over the study region is primarily generated by localized convective instability (due to local surface heating and uplifting) with moisture supply from local sources

(Shrestha et al., 2012) or by the occasional passage of westerly disturbances during early months. However, due to the location of the target area in Eastern Nepal, a substantial amount of precipitation occurs in late pre-monsoon season, when moisture supply from the Bay of Bengal starts to increase

*8. I see most of the local scale features in D2 and D3 are closely resemble with each other. How good is D3 compared to D2? The authors can briefly state this in the conclusions.*

**Response: We have now discussed it in conclusions by adding following lines (at line 23 page 15).**

Although the essential features between both 5 and 1 km convection permitting simulation are the same, the later reduces the overestimated magnitude of precipitation and biases in temperature, and especially better reproduces the timing and magnitude of monsoonal diurnal cycle of precipitation, indicating the dominant influence of topography and surface fields (atmospheric dynamics) on resolving the topo-climates.

*9. Although, it is not in scope of this manuscript, the authors can briefly comment on interaction between the westerlies and monsoonal circulation.*

**Response: We have added following lines in the study area section (line 30 page 4)**

In general, the northward migration of monsoonal trough (break monsoon in main land India) towards Himalaya increases the rainfall activity and intense precipitation events over the region. When these northward migrated monsoonal system further merge with upper level extra-tropical westerly trough (south ward migrated) then it often provides a highly favorable synoptic situation to converge huge moisture from both Arabian sea and Bay of Bengal resulting in catastrophic extreme precipitation events (Shrestha, 2016).

**References:**

- Alexandru, A., de Elia, R., Laprise, R., Separovic, L. and Biner, S.: Sensitivity Study of Regional Climate Model Simulations to Large-Scale Nudging Parameters, Mon. Weather Rev., 137(5), 1666–1686, doi:10.1175/2008MWR2620.1, 2008.
- Collier, E. and Immerzeel, W. W.: High-resolution modeling of atmospheric dynamics in the Nepalese H, J. Geophys. Res. Atmos., 120(19), 9882–9896, doi:10.1002/2015JD023266.Received, 2015.
- Norris, J., Carvalho, L. M. V, Jones, C., Cannon, F., Bookhagen, B., Palazzi, E. and Tahir, A. A.: The spatiotemporal variability of precipitation over the Himalaya: evaluation of one-year WRF model simulation, Clim. Dyn., 1–26, doi:10.1007/s00382-016-3414-y, 2016.
- Otte, T. L., Nolte, C. G., Otte, M. J. and Bowden, J. H.: Does nudging squelch the extremes in regional climate modeling?, J. Clim., 25(20), 7046–7066, doi:10.1175/JCLI-D-12-00048.1, 2012.
- Pohl, B. and Crétat, J.: On the use of nudging techniques for regional climate modeling: application for tropical convection, Clim. Dyn., 43(5), 1693–1714, doi:10.1007/s00382-013-1994-3, 2014.
- Shrestha, A.: Cloudbursts in the Nepal Himalayas : Interaction between the Indian monsoon and extratropics. Ph.D. thesis, University of Wisconsin-Madison, USA, 2016.
- Shrestha, D., Singh, P. and Nakamura, K.: Spatiotemporal variation of rainfall over the central Himalayan region revealed by TRMM Precipitation Radar, J. Geophys. Res. Atmos., 117(22), 1–14, doi:10.1029/2012JD018140, 2012.

- Von Storch, H., Langenberg, H., and Feser, F.: A spectral nudging technique for dynamical downscaling purposes, Mon. Weather Rev., 128(10), 3664–3673, 2000.

---

## Author Comment (AC2) · 23 May 2017

**Reviewer 2:**

We sincerely thank the reviewer for taking the time to thorough review our manuscript and for providing the helpful and constructive comments and suggestions for improvement. We hope that we have addressed the Referee's comments -our responses are in black color font.

*The manuscript's goal to better understand the importance and effects of grid resolution on climate simulation in complex terrain is important. Especially, there is still a lack of studies with regional climate models in the convection permitting scale range. The manuscript is well designed and written in general too. I have two major issues (or better to say: wishes) and a few minor comments as given below. Therefore, I suggest a major revision.*

1. *The manuscript discusses simulations with 25, 5, and 1 km. One result is that the 25-km simulation misses the observed diurnal cycle and the authors explain this by a lack of ability of the simulation to resolve the complex orography. Another explanation might be that convection parameterisations have an issue in simulating a proper diurnal cycle. It would add value to the manuscript if the authors can do simulations with 5 or 1 km set-ups, but the orography smoothed to the 25-km simulation orography. This would strongly support the authors conclusion.*

Response: We agree with the reviewer that convection parameterization in general has a common early convection triggering issue. To address the issue raised, we performed monsoon season simulation for 5km with 10 arc minute (~20 km) topography (abbreviated D2_st) keeping spin up and all other options exactly same. The updated Figure 5 with addition of new D2_st line and new Supplement Figure 2 are added (enclosed herewith) in the revised manuscript. We have also enclosed two more figures from our original manuscript related to these features to make the comparison easier to follow here.

- Based on this additional simulation, we have **also highlighted the impact of convection** permitting option (convection parameterization related deficiencies in sub-grid scale precipitation) wherever its role was not properly mentioned (or shadowed), however dominancy of topography to resolve the timings and shape of diurnal cycle, and spatial distribution of precipitation during monsoon in study area is quite clear and it does not alter our main conclusion on dominancy of topography to resolve these features (Figures below).

  **We have mainly addressed and highlight following issues in the revised manuscript**
  A. The potential deficiencies of convection parameterization in coarser resolution besides the topography for its deficiency on representing spatial distribution of precipitation in monsoon season is mentioned.
  B. It is quite clear from new Supplement Figure 2 that D2 with coarse topography (D2_st) completely misses the **double precipitation maximum zones with the drier valleys in between during monsoon** as evident in Figure 10 (original supplement Figure 2; D2 Monsoon) and widely demonstrated by previous studies (Karki et al., 2017, Bookhagen and Burbank, 2006, Shrestha et al., 2012, Maussian et al., 2014, Gerlitz et al., 2015). However, it follows its own topography showing mountain, valley contrast for precipitation distribution to the scale topography is resolved there. It thus also indicates how strongly the topography affects the precipitation distribution in complex Himalayan terrain.

C. Though, D2_st even with its coarse topography ( ~20 km) resolves the large scale southern foothills mid night to early morning monsoonal precipitation peak features to some extent (but random compared to D2) with the impact of explicitly resolved convection (and as a result of relatively less heterogeneous terrain in south), **the mountain valley contrast in our study area is very unrealistic** compared to D2 (Supplementary Figure 2 and Figure 17 comparison, and updated Figure 5) due to its coarse topography.

Further, the pattern seems unrealistic and shows evenly distributed peak hours from north to south in western side (Supplement Figure 2 and Figure 17).

**Hence, it supports our conclusion of dominancy of topography on resolving diurnal cycle (mountain valley circulation) in the study area.**

We have presented and discussed briefly the results from the additional simulation in our revised manuscript now.

[Figure]

**Updated Figure 5**: Diurnal precipitation during monsoon seasons in different WRF resolutions and observation, categorized into overall (all average), ridge, upper valley and lower valley.

[Figure]

**New planned Supplement Figure 2:** Spatial distribution of a) monsoonal precipitation (mm) and b) peak precipitation hour during monsoon in D2_st domain (shaded). Elevation contour from D2_st topography is also plotted and labelled at every 1000 m.

[Figure]

**Figure 10 in revised manuscript (Supplement Figure 2 in original manuscript) :** Seasonal distribution of precipitation in larger area in D2 and D1 domain resolutions.

[Figure]

**Figure 17 in revised manuscript (original Figure 16):** Spatial pattern of peak precipitation hour (local) during monsoon season in large area of D2 and D1.

2. *There is no shallow convection parameterisation applied in the 5-km set-up? Please, clarify. I assume simulations with or without shallow convection differ substantially in the case of the 5-km set-up.*

Response: We agree with the reviewer that the result can differ as these are the gray zones. However, the multi-year simulation by one of our co-authors (Shabeh ul Hasson) for the whole Himalayan region using KF convection parameterization (resolves shallow convection as well) at 6 km resolution identified the spuriously very high precipitation in random areas (not shown) for some years. In addition, Collier and Immerzeel (2015) also found the further worsening of precipitation output in their short period simulation in Himalaya with inclusion of KF scheme at 5 km.

For these reasons, we had performed 5 km simulation by explicitly resolving convection (not included shallow as well) as simulated earlier for similar grid resolution (Collier and Immerzeel, 2015, Norris et al., 2016). The result from our 5 km simulation has captured the essential features that 1 km shows. It is further consistent to Norris et al., 2016 simulation with explicitly resolved 6.7 km simulation indentifying overall overestimation of the similar magnitude.

Thank you very much if you are directing to use the independent shallow convection parameterization scheme available in WRF are not tested for the region and might perform better. We will definitely consider the limited (2 schemes) independent shallow convection available in WRF and do sensitivity experiments if we consider similar scale in our future simulations.

Given the computational resources limitations to repeat whole simulations, and the time to rewrite and revise the whole manuscript it takes, we keep it as limitation for the current study.

To clarify if shallow convection parameterization is used or not, and to support why we turned off convection parameterization, we have added the following lines in revised manuscript (line 12 page 5).

Although D2 is gray zones for explicitly resolving the convection, Collier and Immerzeel (2015) found the further worsening of precipitation output in their short period simulation in Himalaya with inclusion of KF scheme (resolves shallow convection as well) compared to explicitly resolved convection at that scale. In addition, the multi-year simulation at 6 km with KF parameterization by one of our co-authors (Shabeh ul Hasson) found the spuriously high precipitation in some areas (not shown) for some years. For these reasons, cumulus parameterization is completely turned off in both D2 and D3 in order to explicitly resolve the convective precipitation processes.

Minor comments:

1. *The first part of the title is too general and in the second part the reference of the comparison is missing (perhaps better to replace comparison with evaluation). I suggest the title to be revised.*

Response: We agree and would like to revise the title as below;
- Quantifying the added value of high resolution climate models: A systematic comparison of WRF simulations for complex Himalayan terrain (original)
- **Quantifying the added value of convection-permitting climate simulations in complex terrain: A systematic evaluation of WRF over the Himalaya (revised).**

2. *Please, give a reference to the TREELINE project already in the introduction.*
Response: corrected.

3. *Sec. 2: decrease of the precipitation in the valleys with and because of north-south orientation?*

Response: It may not be due to north-south orientation only because similar decrease is observed in east-west orientated Rolwaling valley. However, the main reason for this is the leeward effect of multiple mountains blocking the main monsoonal flow.

4. *Eq. 1 & 2: Index m missing at the Os in the numerators?*

Response: We used alternative approach taking difference from mean rather than individual observed values for precipitation since the common approach is not meaningful (denominator 0) in the case with 0 precipitation values (Myansbrugge, 2010). Theoretically, the approach is same.

5. *You refer to Fig. S1 two times. If it is an important Fig. think about putting it in the main text.*

Response: Now, realizing the importance of these figure and their mention in several places, we have moved both Figures from supplement to main text in revised manuscript.

6. *Page 9, line 25: These are the mean observed temperatures probably?*
Response: corrected.

7. *At the end of Sec. 4.1 there is some speculation which should be avoided.*
Response: corrected

8.  *Page 12, line 23: Earlier in the manuscript it was mentioned that bias correction was applied for temperature. Why does the coarse simulation show smoothness after bias correction using the fine-scale orography?*

Response: The writing (line 37 page 6) was making confusions to readers. We applied altitude correction only to the data which were extracted for station location but not for all the simulated grid points. We rewrote the sentence now as:

> Thus, simulated temperatures extracted for station locations (not entire grids used for spatial mapping) only are adjusted using a constant lapse rate of 6 $^\circ$Ckm-1, which has been observed over the whole Koshi basin (Salerno et al., 2015).

9.  *Fig. 2 can be omitted. I think the text can be shortened quite a bit if carefully reconsidered. There are some minor language issues like missing "a"s, "s"s and "the"s*

Response: We moved this figure to supplement and deleted most of the contents related to this in main text. The unnecessary texts are further omitted and the grammatical errors tried to reduce.

**References:**

- Bookhagen, B. and Burbank, D. W.: Topography, relief, and TRMM-derived rainfall variations along the Himalaya, Geophys. Res. Lett., 33(8), 1–5, doi:10.1029/2006GL026037, 2006.

- Collier, E. and Immerzeel, W. W.: High-resolution modeling of atmospheric dynamics in the Nepalese H, J. Geophys. Res. Atmos., 120(19), 9882–9896, doi:10.1002/2015JD023266.Received, 2015.

- Gerlitz, L., Conrad, O. and Böhner, J.: Large-scale atmospheric forcing and topographic modification of precipitation rates over High Asia – a neural-network-based approach, Earth Syst. Dyn., 6(1), 61–81, doi:10.5194/esd-6-61-2015, 2015.

- Karki, R., Hasson, S. ul, Schickhoff, U., Scholten, T. and Böhner, J.: Rising Precipitation Extremes across Nepal, Climate, 5(1), 2017.

- Maussion, F., Scherer, D., Mölg, T., Collier, E., Curio, J. and Finkelnburg, R.: Precipitation seasonality and variability over the Tibetan Plateau as resolved by the high Asia reanalysis, J. Clim., 27(5), 1910–1927, doi:10.1175/JCLI-D-13-00282.1, 2014.

- Mynsbrugge, J.V.: Bidding Strategies Using Price Based Unit Commitment in a Deregulated Power Market, K.U.Leuven, 2010 (http://en.wikipedia.org/wiki/Mean_absolute_percentage_error , accessed 11.5.2017)

- Norris, J., Carvalho, L. M. V, Jones, C., Cannon, F., Bookhagen, B., Palazzi, E. and Tahir, A. A.: The spatiotemporal variability of precipitation over the Himalaya: evaluation of one-year WRF model simulation, Clim. Dyn., 1–26, doi:10.1007/s00382-016-3414-y, 2016.

- Shrestha, D., Singh, P. and Nakamura, K.: Spatiotemporal variation of rainfall over the central Himalayan region revealed by TRMM Precipitation Radar, J. Geophys. Res. Atmos., 117(22), 1–14, doi:10.1029/2012JD018140, 2012.